# Mixed-Type Tabular Data Synthesis with Score-based Diffusion in Latent Space

**Hengrui Zhang[1]*  Jiani Zhang[2]†  Balasubramaniam Srinivasan[2]  Zhengyuan Shen[2]**
**Xiao Qin[2]  Christos Faloutsos[2]  Huzefa Rangwala[2,3]‡  George Karypis[2]**
[1]Computer Science Department, University of Illinois at Chicago    [2]Amazon Web Services
[3]Computer Science, George Mason University
hzhan55@uic.edu    {zhajiani,srbalasu,donshen}@amazon.com
{drxqin,faloutso,rhuzefa,gkarypis}@amazon.com

## Abstract

Recent advances in tabular data generation have greatly enhanced synthetic data quality. However, extending diffusion models to tabular data is challenging due to the intricately varied distributions and a blend of data types of tabular data. This paper introduces TABSYN, a methodology that synthesizes tabular data by leveraging a diffusion model within a variational autoencoder (VAE) crafted latent space. The key advantages of the proposed TABSYN include (1) **Generality**: the ability to handle a broad spectrum of data types by converting them into a single unified space and explicitly capture inter-column relations, (2) **Quality**: optimizing the distribution of latent embeddings to enhance the subsequent training of diffusion models, which helps generate high-quality synthetic data, (3) **Speed**: much fewer number of reverse steps and faster synthesis speed than existing diffusion-based methods. Extensive experiments on six datasets with five metrics demonstrate that TABSYN outperforms existing methods. Specifically, it reduces the error rates by *86%* and *67%* for column-wise distribution and pair-wise column correlation estimations compared with the most competitive baselines. Code has been made available at https://github.com/amazon-science/tabsyn.

## 1 Introduction

Tabular data synthesis has a wide range of applications, such as augmenting training data (Fonseca & Bacao, 2023), protecting private data instances (Assefa et al., 2021; Hernandez et al., 2022), and imputing missing values (Zheng & Charoenphakdee, 2022). Recent developments in tabular data generation have notably enhanced the quality of synthetic data (Xu et al., 2019; Borisov et al., 2023; Liu et al., 2023b), while the synthetic data is still far from the real one. To further improve the generation quality, researchers have explored adapting diffusion models, which have shown strong performance in image synthesis tasks (Ho et al., 2020; Rombach et al., 2022),

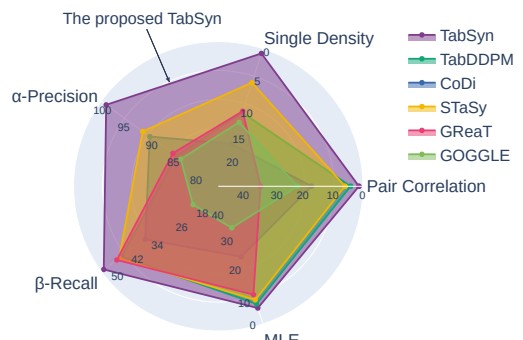

Figure 1: Our TABSYN consistently outperforms SOTA tabular data generation methods across five data quality metrics.

for tabular data generation (Kim et al., 2022; Kotelnikov et al., 2023; Kim et al., 2023; Lee et al., 2023). Despite the progress made by these methods, tailoring a diffusion model for tabular data leads to several challenges. Unlike image data, which comprises pure continuous pixel values with local spatial correlations, tabular data features have complex and varied distributions (Xu et al.,

*Work conducted during an internship at Amazon Web Services.
†Corresponding author.
‡Huzefa Rangwala is on LOA as a Professor of Computer Science at George Mason University. This paper describes work performed at Amazon.

2019), making it hard to learn joint probabilities across multiple columns. Moreover, typical tabular data often contains mixed data types, i.e., continuous (e.g., numerical features) and discrete (e.g., categorical features) variables. The standard diffusion process assumes a continuous input space with Gaussian noise perturbation, which leads to additional challenges with categorical features. Existing solutions either transform categorical features into numerical ones using techniques like one-hot encoding (Kim et al., 2023; Liu et al., 2023b) and analog bit encoding (Zheng & Charoenphakdee, 2022) or resort to two separate diffusion processes for numerical and categorical features (Kotelnikov et al., 2023; Lee et al., 2023). However, it has been proven that simple encoding methods lead to suboptimal performance (Lee et al., 2023), and learning separate models for different data types makes it challenging for the model to capture the co-occurrence patterns of different types of data. Therefore, we seek to develop a diffusion model in a joint space of numerical and categorical features that preserves the inter-column correlations.

This paper presents TABSYN, a principled approach for tabular data synthesis. To handle mixed-typed inputs, TABSYN first transforms raw tabular data into a continuous embedding space, where well-developed diffusion models with Gaussian noises become feasible. Subsequently, we learn a score-based diffusion model in the embedding space to capture the distribution of latent embeddings. To learn an informative, smoothed latent space while maintaining the decoder's reconstruction ability, we specifically designed a Variational AutoEncoder (VAE (Kingma & Welling, 2013)) model for tabular-structured data. Our proposed VAE model includes 1) Transformer-architecture encoders and decoders for modeling inter-column relationships and obtaining token-level representations, facilitating token-level tasks. 2) Adaptive loss weighting to dynamically adjust the reconstruction loss weights and KL-divergence weights, allowing the model to improve reconstruction performance gradually while maintaining a regularized embedding space. 3) Finally, when applying diffusion models in the latent space, we adopt a simplified forward diffusion process, which adds Gaussian noises of linear standard deviation with respect to time. We demonstrate through theoretical analysis and empirical justifications that this approach can reduce the errors in the reverse process, thus improving sampling speed.

The advantages of TABSYN are three-fold: (1) **Generality**: *Mixed-type Feature Handling* - TABSYN transforms diverse input features, encompassing numerical, categorical, etc., into a unified embedding space. (2) **Quality**: *High Generation Quality* - with tailored designs of the VAE model, the tabular data is mapped into regularized latent space of good shape, e.g., a standard normal distribution. This will greatly simplify training the subsequent diffusion model (Vahdat et al., 2021), making TABSYN more expressive and enabling it to generate high-quality synthetic data. (3) **Speed**: With the proposed linear noise schedule, our TABSYN can generate high-quality synthetic data with fewer than 20 reverse steps, which is significantly fewer than existing methods.

Recognizing the absence of unified and comprehensive evaluations for synthetic tabular data (Du & Li, 2024), we perform extensive experiments, which involve comparing TABSYN with seven state-of-the-art methods on six mixed-type tabular datasets using over five distinct evaluation metrics. The experimental results demonstrate that TABSYN consistently outperforms previous methods (see Figure 1). Specifically, TABSYN reduces the average errors in column-wise distribution shape estimation (i.e., single density) and pair-wise column correlation estimation (i.e., pair correlation) tasks by $86\%$ and $67\%$ than the most competitive baselines. Furthermore, we demonstrate that TABSYN achieves competitive performance across two downstream tabular data tasks, machine learning efficiency and missing value imputation. Specifically, the well-learned unconditional TABSYN is able to be applied to missing value imputation without retraining. Moreover, thorough ablation studies and visualization case studies substantiate the rationale and effectiveness of our developed approach.

## 2  RELATED WORKS

**Deep Generative Models for Tabular Data Generation.**   Generative models for tabular data have become increasingly important and have widespread applications Assefa et al. (2021); Zheng & Charoenphakdee (2022); Hernandez et al. (2022). To deal with the imbalanced categorical features, Xu et al. (2019) proposes CTGAN and TVAE based on the popular Generative Adversarial Networks (Goodfellow et al., 2014) and VAE (Kingma & Welling, 2013), respectively. Multiple advanced methods have been proposed for synthetic tabular data generation in the past year. Specifically, GOG-GLE (Liu et al., 2023b) became the first to explicitly model the dependency relationship between

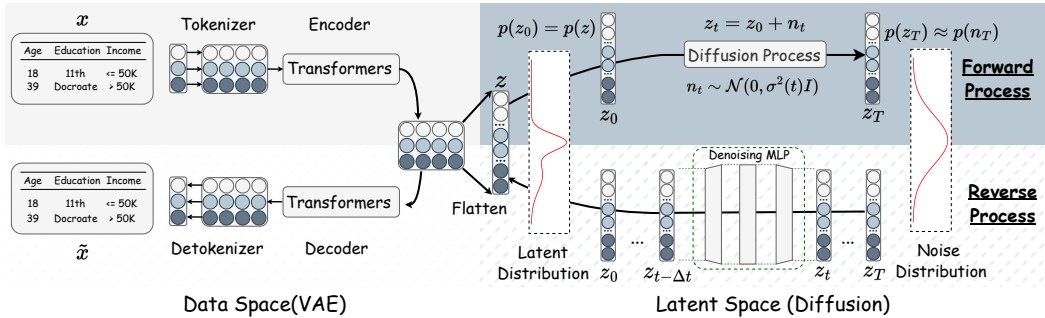

Figure 2: An overview of the proposed TABSYN. Each row data $x$ is mapped to latent space $z$ via a column-wise tokenizer and an encoder. A diffusion process $z_0 \rightarrow z_T$ is applied in the latent space. Synthesis $z_T \rightarrow z_0$ starts from the base distribution $p(z_T)$ and generates samples $z_0$ in latent space through a reverse process. These samples are then mapped from latent $z$ to data space $\tilde{x}$ using a decoder and a detokenizer.

columns, proposing a VAE-based model using graph neural networks as the encoder and decoder models. Inspired by the success of large language models in modeling the distribution of natural languages, GReaT transformed each row in a table into a natural sentence and learned sentence-level distributions using auto-regressive GPT2. In recent years, the physical diffusion process has inspired a lot of advanced research in deep learning. For example, DIFFormer (Wu et al., 2023) develops a scalable Transformer model for geometric data via a constrained diffusion process, and the Denoising Diffusion models have achieved great success in image generation (Ho et al., 2020). STaSy (Kim et al., 2023), TabDDPM (Kotelnikov et al., 2023), and CoDi (Lee et al., 2023) concurrently applied the popular diffusion-based generative models for synthetic tabular data generation.

**Generative Modeling in the Latent Space.** While generative models in the data space have achieved significant success, latent generative models have demonstrated several advantages, including more compact and disentangled representations, robustness to noise, and greater flexibility in controlling generated styles (van den Oord et al., 2017; Razavi et al., 2019; Esser et al., 2021). For example, the recent GAN literature (Li et al., 2022) has demonstrated superior controllability via adversarial learning in the latent space. Recently, the Latent Diffusion Models (LDM) (Rombach et al., 2022; Vahdat et al., 2021) have achieved great success in image generation as they exhibit better scaling properties and expressivity than the vanilla diffusion models in the data space (Ho et al., 2020; Song et al., 2021b; Karras et al., 2022). The success of LDMs in image generation has also inspired their applications in video (Blattmann et al., 2023) and audio data (Liu et al., 2023a). To the best of our knowledge, the proposed work is the first to explore the application of latent diffusion models for general tabular data generation tasks.

## 3 SYNTHETIC TABULAR DATA GENERATION WITH TABSYN

Figure 2 gives an overview of TABSYN. In Section 3.1, we first formally define the tabular data generation task. Then, we introduce the design details of TABSYN's autoencoding and diffusion process in Section 3.2 and 3.3. We summarize the training and sampling algorithms in Appendix A.

### 3.1 PROBLEM DEFINITION OF TABULAR DATA GENERATION

Let $M_{\text{num}}$ and $M_{\text{cat}}$ be the number of numerical columns and categorical columns, respectively. Each row is represented as a vector of numerical features and categorical features $\boldsymbol{x} = [\boldsymbol{x}^{\text{num}}, \boldsymbol{x}^{\text{cat}}]$, where $\boldsymbol{x}^{\text{num}} \in \mathbb{R}^{M_{\text{num}}}$ and $\boldsymbol{x}^{\text{cat}} \in \mathbb{R}^{M_{\text{cat}}}$. Specifically, the $i$-th categorical attribute has $C_i$ finite candidate values, therefore we have $x_i^{\text{cat}} \in \{1, \cdots, C_i\}, \forall i$. This paper focuses on the **unconditional generation** task. With a tabular dataset $\mathcal{T} = \{\boldsymbol{x}\}$, we aim to learn a parameterized generative model $p_\theta(\mathcal{T})$, with which realistic and diverse synthetic tabular data $\hat{\boldsymbol{x}} \in \hat{\mathcal{T}}$ can be generated.

## 3.2 AUTOENCODING FOR TABULAR DATA

Tabular data is highly structured of mixed-type column features, with different columns having distinct meanings and being highly dependent on each other. These characteristics make it challenging to design an approximate encoder to model and effectively utilize the rich relationships between columns. Motivated by the successes of Transformers in classification/regression of tabular data (Gorishniy et al., 2021), we first learn a unique tokenizer for each column, and then the token(column)-wise representations are fed into a Transformer for capturing the intricate relationships among columns.

**Feature Tokenizer**. The feature tokenizer converts each column (both numerical and categorical) into a $d$-dimensional vector. First, we use one-hot encoding to pre-process categorical features, i.e., $x_i^{\text{cat}} \Rightarrow \boldsymbol{x}_i^{\text{oh}} \in \mathbb{R}^{1 \times C_i}$. Each record is represented as $\boldsymbol{x} = [\boldsymbol{x}^{\text{num}}, \boldsymbol{x}_1^{\text{oh}}, \cdots, \boldsymbol{x}_{M_{\text{cat}}}^{\text{oh}}] \in \mathbb{R}^{M_{\text{num}} + \sum_{i=1}^{M_{\text{cat}}} C_i}$. Then, we apply a linear transformation for numerical columns and create an embedding lookup table for categorical columns, where each category is assigned a learnable $d$-dimensional vector, i.e.,

$$\boldsymbol{e}_i^{\text{num}} = x_i^{\text{num}} \cdot \boldsymbol{w}_i^{\text{num}} + \boldsymbol{b}_i^{\text{num}}, \quad \boldsymbol{e}_i^{\text{cat}} = \boldsymbol{x}_i^{\text{oh}} \cdot \boldsymbol{W}_i^{\text{cat}} + \boldsymbol{b}_i^{\text{cat}}, \tag{1}$$

where $\boldsymbol{w}_i^{\text{num}}, \boldsymbol{b}_i^{\text{num}}, \boldsymbol{b}_i^{\text{cat}} \in \mathbb{R}^{1 \times d}$, $\boldsymbol{W}_i^{\text{cat}} \in \mathbb{R}^{C_i \times d}$ are learnable parameters of the tokenizer, $\boldsymbol{e}_i^{\text{num}}, \boldsymbol{e}_i^{\text{cat}} \in \mathbb{R}^{1 \times d}$. Now, each record is expressed as the stack of the embeddings of all columns

$$\boldsymbol{E} = [\boldsymbol{e}_1^{\text{num}}, \cdots, \boldsymbol{e}_{M_{\text{num}}}^{\text{num}}, \boldsymbol{e}_1^{\text{cat}}, \cdots, \boldsymbol{e}_{M_{\text{cat}}}^{\text{cat}}] \in \mathbb{R}^{M \times d}. \tag{2}$$

**Transformer Encoding and Decoding**. As with typical VAEs, we use the encoder to obtain the mean and log variance of the latent variable. Then, we acquire the latent embeddings with the reparameterization tricks. The latent embeddings are then passed through the decoder to obtain the reconstructed token matrix $\hat{\boldsymbol{E}} \in \mathbb{R}^{M \times d}$. The detailed architectures are in Appendix D.

**Detokenizer**. Finally, we apply a detokenizer to the recovered token representation of each column to reconstruct the column values. The design of the detokenizer is symmetrical to that of the tokenizer:

$$\hat{x}_i^{\text{num}} = \hat{\boldsymbol{e}}_i^{\text{num}} \cdot \hat{\boldsymbol{w}}_i^{\text{num}} + \hat{b}_i^{\text{num}}, \quad \hat{\boldsymbol{x}}_i^{\text{oh}} = \text{Softmax}(\hat{\boldsymbol{e}}_i^{\text{cat}} \cdot \hat{\boldsymbol{W}}_i^{\text{cat}} + \hat{\boldsymbol{b}}_i^{\text{cat}}),$$
$$\hat{\boldsymbol{x}} = [\hat{x}_1^{\text{num}}, \cdots, \hat{x}_{M_{\text{num}}}^{\text{num}}, \hat{\boldsymbol{x}}_1^{\text{oh}}, \cdots, \hat{\boldsymbol{x}}_{M_{\text{cat}}}^{\text{oh}}], \tag{3}$$

where $\hat{\boldsymbol{w}}_i^{\text{num}} \in \mathbb{R}^{d \times 1}, \hat{b}_i^{\text{num}} \in \mathbb{R}^{1 \times 1}, \boldsymbol{W}_i^{\text{cat}} \in \mathbb{R}^{d \times C_i}, \hat{\boldsymbol{b}}_i^{\text{cat}} \in \mathbb{R}^{1 \times C_i}$ are detokenizer's parameters.

**Training with adaptive weight coefficient.** The VAE model is usually learned with the classical ELBO loss function, but here we use $\beta$-VAE (Higgins et al., 2016), where a coefficient $\beta$ balances the importance of the reconstruction loss and KL-divergence loss

$$\mathcal{L} = \ell_{\text{recon}}(\boldsymbol{x}, \hat{\boldsymbol{x}}) + \beta \ell_{\text{kl}}. \tag{4}$$

$\ell_{\text{recon}}$ is the reconstruction loss between the input data and the reconstructed one, and $\ell_{\text{kl}}$ is the KL divergence loss that regularizes the mean and variance of the latent space. In the vanilla VAE model, $\beta$ is set to be 1 because the two loss terms are equally important to generate high-quality synthetic data from Gaussian noises. However, in our model, $\beta$ is expected to be smaller, as we do not require the distribution of the embeddings to precisely follow a standard Gaussian distribution because we have an additional diffusion model. Therefore, we propose to adaptively schedule the scale of $\beta$ in the training process, encouraging the model to achieve lower reconstruction error while maintaining an appropriate embedding shape.

With an initial (maximum) $\beta = \beta_{\max}$, we monitor the epoch-wise reconstruction loss $\ell_{\text{recon}}$. When $\ell_{\text{recon}}$ fails to decrease for a predefined number of epochs (which indicates that the KL-divergence dominates the overall loss), the weight is scheduled by $\beta = \lambda\beta, \lambda < 1$. This process continues until $\beta$ approaches a predefined minimum value $\beta_{\min}$. This strategy is simple yet very effective, and we empirically justify the effectiveness of the design in Section 4.

## 3.3 SCORE-BASED GENERATIVE MODELING IN THE LATENT SPACE

**Training and sampling via denoising.** After the VAE model is well-learned, we extract the latent embeddings through the encoder and flatten the encoder's output as $\boldsymbol{z} = \text{Flatten}(\text{Encoder}(\boldsymbol{x})) \in \mathbb{R}^{1 \times Md}$ such that the embedding of a record is a vector rather than a matrix. To learn the underlying

distribution of embeddings $p(\boldsymbol{z})$, we consider the following forward diffusion process and reverse sampling process (Song et al., 2021b; Karras et al., 2022):

$$\boldsymbol{z}_t = \boldsymbol{z}_0 + \sigma(t)\boldsymbol{\varepsilon}, \ \boldsymbol{\varepsilon} \sim \mathcal{N}(\mathbf{0}, \boldsymbol{I}), \qquad\qquad \text{(Forward Process)} \qquad (5)$$

$$\mathrm{d}\boldsymbol{z}_t = -2\dot{\sigma}(t)\sigma(t)\nabla_{\boldsymbol{z}_t} \log p(\boldsymbol{z}_t)\mathrm{d}t + \sqrt{2\dot{\sigma}(t)\sigma(t)}\mathrm{d}\boldsymbol{\omega}_t, \qquad \text{(Reverse Process)} \qquad (6)$$

where $\boldsymbol{z}_0 = \boldsymbol{z}$ is the initial embedding from the encoder, $\boldsymbol{z}_t$ is the diffused embedding at time $t$, and $\sigma(t)$ is the noise level. In the reverse process, $\nabla_{\boldsymbol{z}_t} \log p_t(\boldsymbol{z}_t)$ is the score function of $\boldsymbol{z}_t$, and $\boldsymbol{\omega}_t$ is the standard Wiener process. The training of the diffusion model is achieved via denoising score matching (Karras et al., 2022):

$$\mathcal{L} = \mathbb{E}_{\boldsymbol{z}_0 \sim p(\boldsymbol{z}_0)}\mathbb{E}_{t \sim p(t)}\mathbb{E}_{\boldsymbol{\varepsilon} \sim \mathcal{N}(\mathbf{0}, \boldsymbol{I})}\|\boldsymbol{\epsilon}_\theta(\boldsymbol{z}_t, t) - \boldsymbol{\varepsilon}\|_2^2, \ \text{ where } \boldsymbol{z}_t = \boldsymbol{z}_0 + \sigma(t)\boldsymbol{\varepsilon}, \qquad (7)$$

where $\boldsymbol{\epsilon}_\theta$ is a neural network (named denoising function) to approximate the Gaussian noise using the perturbed data $\boldsymbol{x}_t$ and the time $t$. Then $\nabla_{\boldsymbol{z}_t} \log p(\boldsymbol{z}_t) = -\boldsymbol{\epsilon}_\theta(\boldsymbol{z}_t, t)/\sigma(t)$. After the model is trained, synthetic data can be obtained via the reverse process in Eq. 6. The detailed algorithm description of TABSYN is provided in Appendix A. Detailed derivations are in Appendix B.

**Schedule of noise level** $\sigma(t)$. The noise level $\sigma(t)$ defines the scale of noises for perturbing the data at different time steps and significantly affects the final Differential Equation solution trajectories (Song et al., 2021b; Karras et al., 2022). Following the recommendations in Karras et al. (2022), we set the noise level $\sigma(t) = t$ that is linear w.r.t. the time. We show in Proposition 1 that the linear noise level schedule leads to the smallest approximation errors in the reverse process:

**Proposition 1.** *Consider the reverse diffusion process in Equation (6) from $\boldsymbol{z}_{t_b}$ to $\boldsymbol{z}_{t_a}(t_b > t_a)$, the numerical solution $\hat{\boldsymbol{z}}_{t_a}$ has the smallest approximation error to $\boldsymbol{z}_{t_a}$ when $\sigma(t) = t$.*

See proof in Appendix C. A natural corollary of Proposition 1 is that a small approximation error allows us to increase the interval between two timesteps, thereby reducing the overall number of sampling steps and accelerating the sampling. In Section 4, we demonstrate that with this design, TABSYN can generate synthetic tabular data of high quality within less than 20 NFEs (number of function evaluations), which is much smaller than other tabular-data synthesis methods based on diffusion (Kim et al., 2023; Kotelnikov et al., 2023).

## 4 BENCHMARKING SYNTHETIC TABULAR DATA GENERATION ALGORITHMS

### 4.1 EXPERIMENTAL SETUPS

**Datasets**. We select six real-world tabular datasets consisting of both numerical and categorical attributes: Adult, Default, Shoppers, Magic, Faults, Beijing, and News. Table 6 provides the overall statistics of these datasets, and the detailed descriptions can be found in Appendix E.1.

**Baselines**. We compare the proposed TABSYN with seven existing synthetic tabular data generation methods. The first two are classical GAN and VAE models: CTGAN (Xu et al., 2019) and TVAE (Xu et al., 2019). Additionally, we evaluate five SOTA methods introduced recently: GOGGLE (Liu et al., 2023b), a VAE-based method; GReaT (Borisov et al., 2023), a language model variant; and three diffusion-based methods: STaSy (Kim et al., 2023), TabDDPM (Kotelnikov et al., 2023), and CoDi (Lee et al., 2023). Notably, these approaches were nearly simultaneously introduced, limiting opportunities for extensive comparison. For reference, we also compare with the representative interpolation-based method SMOTE (Chawla et al., 2002). Our paper fills this gap by providing the first comprehensive evaluation of their performance in a standardized setting.

**Evaluation Methods**. We evaluate the quality of the synthetic data from three aspects: 1) **Low-order statistics** – *column-wise density estimation* and *pair-wise column correlation*, estimating the density of every single column and the correlation between every column pair (Section 4.2). We also perform Classifier Two Sample Test (C2ST) to evaluate if the synthetic data can be detected from the real data via a machine learning model (Appendix F.3); 2) **High-order metrics** – $\alpha$-*precision* and $\beta$-*recall* scores (Alaa et al., 2022) that measure the overall fidelity and diversity of synthetic data (the results are deferred to Appendix F.2); 3) **Privacy Protection**: we also evaluate if the synthetic data is randomly sampled according to the distribution density rather than copied from the training data via Distance to Closest Records (DCR) in Appendix F.6; 4) Performance on **downstream tasks** –

Table 1: Error rate (%) of **column-wise density estimation**. **Bold Face** represents the best score on each dataset. Lower values indicate more accurate estimation (superior results). TABSYN outperforms the best generative baseline model by $86.0\%$ on average.

| Method | Adult | Default | Shoppers | Magic | Beijing | News | Average |
|---|---|---|---|---|---|---|---|
| SMOTE | $1.60_{\pm 0.23}$ | $1.48_{\pm 0.15}$ | $2.68_{\pm 0.19}$ | $0.91_{\pm 0.05}$ | $1.85_{\pm 0.21}$ | $5.31_{\pm 0.46}$ | 2.30 |
| CTGAN | $16.84_{\pm 0.03}$ | $16.83_{\pm 0.04}$ | $21.15_{\pm 0.10}$ | $9.81_{\pm 0.08}$ | $21.39_{\pm 0.05}$ | $16.09_{\pm 0.02}$ | 17.02 |
| TVAE | $14.22_{\pm 0.08}$ | $10.17_{\pm 0.05}$ | $24.51_{\pm 0.06}$ | $8.25_{\pm 0.06}$ | $19.16_{\pm 0.06}$ | $16.62_{\pm 0.03}$ | 15.49 |
| GOGGLE[1] | 16.97 | 17.02 | 22.33 | 1.90 | 16.93 | 25.32 | 16.74 |
| GReaT[2] | $12.12_{\pm 0.04}$ | $19.94_{\pm 0.06}$ | $14.51_{\pm 0.12}$ | $16.16_{\pm 0.09}$ | $8.25_{\pm 0.12}$ | OOM | 14.20 |
| STaSy | $11.29_{\pm 0.06}$ | $5.77_{\pm 0.06}$ | $9.37_{\pm 0.09}$ | $6.29_{\pm 0.13}$ | $6.71_{\pm 0.03}$ | $6.89_{\pm 0.03}$ | 7.72 |
| CoDi | $21.38_{\pm 0.06}$ | $15.77_{\pm 0.07}$ | $31.84_{\pm 0.05}$ | $11.56_{\pm 0.26}$ | $16.94_{\pm 0.02}$ | $32.27_{\pm 0.04}$ | 21.63 |
| TabDDPM[3] | $1.75_{\pm 0.03}$ | $1.57_{\pm 0.08}$ | $2.72_{\pm 0.13}$ | $1.01_{\pm 0.09}$ | $1.30_{\pm 0.03}$ | $78.75_{\pm 0.01}$ | 14.52 |
| TABSYN | $\mathbf{0.58_{\pm 0.06}}$ | $\mathbf{0.85_{\pm 0.04}}$ | $\mathbf{1.43_{\pm 0.24}}$ | $\mathbf{0.88_{\pm 0.09}}$ | $\mathbf{1.12_{\pm 0.05}}$ | $\mathbf{1.64_{\pm 0.04}}$ | **1.08** |
| Improv. | $\mathbf{66.9\% \downarrow}$ | $\mathbf{45.9\% \downarrow}$ | $\mathbf{47.4\% \downarrow}$ | $\mathbf{12.9\% \downarrow}$ | $\mathbf{13.8\% \downarrow}$ | $\mathbf{76.2\% \downarrow}$ | $\mathbf{86.0\% \downarrow}$ |

[1] GOGGLE fixes the random seed during sampling in the official codes, and we follow it for consistency.
[2] GReaT cannot be applied on News because of the maximum length limit.
[3] TabDDPM fails to generate meaningful content on the News dataset.

Table 2: Error rate (%) of **pair-wise column** correlation score. **Bold Face** represents the best score on each dataset. TABSYN outperforms the best baseline model by $67.6\%$ on average.

| Method | Adult | Default | Shoppers | Magic | Beijing | News | Average |
|---|---|---|---|---|---|---|---|
| SMOTE | $3.28_{\pm 0.29}$ | $8.41_{\pm 0.38}$ | $3.56_{\pm 0.22}$ | $3.16_{\pm 0.41}$ | $2.39_{\pm 0.35}$ | $5.38_{\pm 0.76}$ | 4.36 |
| CTGAN | $20.23_{\pm 1.20}$ | $26.95_{\pm 0.93}$ | $13.08_{\pm 0.16}$ | $7.00_{\pm 0.19}$ | $22.95_{\pm 0.08}$ | $5.37_{\pm 0.05}$ | 15.93 |
| TVAE | $14.15_{\pm 0.88}$ | $19.50_{\pm 0.95}$ | $18.67_{\pm 0.38}$ | $5.82_{\pm 0.49}$ | $18.01_{\pm 0.08}$ | $6.17_{\pm 0.09}$ | 13.72 |
| GOGGLE | 45.29 | 21.94 | 23.90 | 9.47 | 45.94 | 23.19 | 28.28 |
| GReaT | $17.59_{\pm 0.22}$ | $70.02_{\pm 0.12}$ | $45.16_{\pm 0.18}$ | $10.23_{\pm 0.40}$ | $59.60_{\pm 0.55}$ | OOM | 44.24 |
| STaSy | $14.51_{\pm 0.25}$ | $5.96_{\pm 0.26}$ | $8.49_{\pm 0.15}$ | $6.61_{\pm 0.53}$ | $8.00_{\pm 0.10}$ | $3.07_{\pm 0.04}$ | 7.77 |
| CoDi | $22.49_{\pm 0.08}$ | $68.41_{\pm 0.05}$ | $17.78_{\pm 0.11}$ | $6.53_{\pm 0.25}$ | $7.07_{\pm 0.15}$ | $11.10_{\pm 0.01}$ | 22.23 |
| TabDDPM | $3.01_{\pm 0.25}$ | $4.89_{\pm 0.10}$ | $6.61_{\pm 0.16}$ | $1.70_{\pm 0.22}$ | $2.71_{\pm 0.09}$ | $13.16_{\pm 0.11}$ | 5.34 |
| TABSYN | $\mathbf{1.54_{\pm 0.27}}$ | $\mathbf{2.05_{\pm 0.12}}$ | $\mathbf{2.07_{\pm 0.21}}$ | $\mathbf{1.06_{\pm 0.31}}$ | $\mathbf{2.24_{\pm 0.28}}$ | $\mathbf{1.44_{\pm 0.03}}$ | **1.73** |
| Improve. | $\mathbf{48.8\% \downarrow}$ | $\mathbf{58.1\% \downarrow}$ | $\mathbf{68.7\% \downarrow}$ | $\mathbf{37.6\% \downarrow}$ | $\mathbf{17.3\% \downarrow}$ | $\mathbf{53.1\% \downarrow}$ | $\mathbf{67.6\% \downarrow}$ |

*machine learning efficiency* (MLE) and *missing value imputation*. MLE is to compare the testing accuracy on real data when trained on synthetically generated tabular datasets. The performance on privacy protection is measured by MLE tasks that have been widely adopted in previous literature (Section 4.3). We also extend TABSYN for the missing value imputation task, which aims to fill in missing features/labels given partial column values (Appendix F.4). The reported results are averaged over 20 randomly sampled synthetic data. The implementation details are in Appendix E.

## 4.2 ESTIMATING LOW-ORDER STATISTICS OF DATA DENSITY

**Metrics**. We employ the Kolmogorov-Sirnov Test (KST) for numerical columns and the Total Variation Distance (TVD) for categorical columns to quantify column-wise density estimation. For pair-wise column correlation, we use Pearson correlation for numerical columns and contingency similarity for categorical columns. The performance is measured by the difference between the correlations computed from real data and synthetic data. For the correlation between numerical and categorical columns, we first group numerical values into categorical ones via bucketing, then calculate the corresponding contingency similarity. Further details on these metrics are in Appendix E.3.

**Column-wise distribution density estimation.** In Table 1, we note that TABSYN consistently outperforms baseline methods in the column-wise distribution density estimation task. On average, TABSYN surpasses the most competitive baselines by $86.0\%$. While STaSy and TabDDPM perform well, STaSy is sub-optimal because it treats one-hot embeddings of categorical columns as continuous features. Additionally, TabDDPM exhibits unstable performance across datasets, failing to generate meaningful content on the News dataset despite a standard training process.

Table 3: AUC (classification task) and RMSE (regression task) scores of **Machine Learning Efficiency**. ↑ (↓) indicates that the higher (lower) the score, the better the performance. TABSYN consistently outperforms all others across all datasets.

| Methods | Adult | Default | Shoppers | Magic | Beijing | News[1] | Average Gap |
|---|---|---|---|---|---|---|---|
| | AUC ↑ | AUC ↑ | AUC ↑ | AUC ↑ | RMSE ↓ | RMSE ↓ | % |
| Real | $.927_{\pm.000}$ | $.770_{\pm.005}$ | $.926_{\pm.001}$ | $.946_{\pm.001}$ | $.423_{\pm.003}$ | $.842_{\pm.002}$ | 0% |
| SMOTE | $.899_{\pm.007}$ | $.741_{\pm.009}$ | $.911_{\pm.012}$ | $.934_{\pm.008}$ | $.593_{\pm.011}$ | $.897_{\pm.036}$ | 9.39% |
| CTGAN | $.886_{\pm.002}$ | $.696_{\pm.005}$ | $.875_{\pm.009}$ | $.855_{\pm.006}$ | $.902_{\pm.019}$ | $.880_{\pm.016}$ | 24.5% |
| TVAE | $.878_{\pm.004}$ | $.724_{\pm.005}$ | $.871_{\pm.006}$ | $.887_{\pm.003}$ | $.770_{\pm.011}$ | $1.01_{\pm.016}$ | 20.9% |
| GOGGLE | $.778_{\pm.012}$ | $.584_{\pm.005}$ | $.658_{\pm.052}$ | $.654_{\pm.024}$ | $1.09_{\pm.025}$ | $.877_{\pm.002}$ | 43.6% |
| GReaT | $.913_{\pm.003}$ | $.755_{\pm.006}$ | $.902_{\pm.005}$ | $.888_{\pm.008}$ | $.653_{\pm.013}$ | OOM | 13.3% |
| STaSy | $.906_{\pm.001}$ | $.752_{\pm.006}$ | $.914_{\pm.005}$ | $.934_{\pm.003}$ | $.656_{\pm.014}$ | $.871_{\pm.002}$ | 10.9% |
| CoDi | $.871_{\pm.006}$ | $.525_{\pm.006}$ | $.865_{\pm.006}$ | $.932_{\pm.003}$ | $.818_{\pm.021}$ | $1.21_{\pm.005}$ | 30.5% |
| TabDDPM[2] | $.907_{\pm.001}$ | $.758_{\pm.004}$ | $.918_{\pm.005}$ | $.935_{\pm.003}$ | $.592_{\pm.011}$ | $4.86_{\pm3.04}$ | 9.14%[1] |
| TABSYN | $.915_{\pm.002}$ | $.764_{\pm.004}$ | $.920_{\pm.005}$ | $.938_{\pm.002}$ | $.582_{\pm.008}$ | $.861_{\pm.027}$ | **7.23%** |

[1] Following CoDi (Lee et al., 2023), the continuous targets are standardized to prevent large values.
[2] TabDDPM collapses on News, leading to an extremely high error on this dataset. We exclude this dataset when computing the average gap of TabDDPM.

**Pair-wise column correlations.**  Table 2 displays the results of pair-wise column correlations. TABSYN outperforms the best baselines by an average of 67.6%. Notably, the performance of GReaT is significantly poorer in this task than in the column-wise task. This indicates the limitations of autoregressive language models in density estimation, particularly in capturing the joint probability distributions between columns.

## 4.3 PERFORMANCE ON DOWNSTREAM TASKS

**Machine Learning Efficiency.**  We then evaluate the quality of synthetic data by evaluating their performance in Machine Learning Efficiency tasks. Following established settings (Kotelnikov et al., 2023; Kim et al., 2023; Lee et al., 2023), we first split a real table into a real training and a real testing set. The generative models are learned on the real training set, from which a synthetic set of equivalent size is sampled. This synthetic data is then used to train a classification/regression model (XGBoost Classifier and XGBoost Regressor (Chen & Guestrin, 2016)), which will be evaluated using the real testing set. The performance of MLE is measured by the AUC score for classification tasks and RMSE for regression tasks. The detailed settings of the MLE evaluations are in Appendix E.4.

In Table 3, we demonstrate that TABSYN consistently outperforms all the baseline methods. The performance gap between methods is smaller compared to column-wise density and pair-wise column correlation estimation tasks (Tables 1 and 2). This suggests that some columns may not significantly impact the classification/regression tasks, allowing methods with lower performance in previous tasks to show competitive results in MLE (e.g., GReaT on Default dataset). This underscores the need for a comprehensive evaluation approach beyond just MLE metrics. As shown above, we have incorporated low-order and high-order statistics for a more robust assessment.

**Missing Value Imputation.**  One advantage of the diffusion model is that a well-trained unconditional model can be directly used for data imputation (e.g., image inpainting (Song et al., 2021b; Lugmayr et al., 2022)) without additional training. This paper explores adapting TABSYN for missing value imputation, a crucial task in real-world tabular data. Due to space limitation, the detailed algorithms for missing value imputation and the results are deferred to Appendix F.4.

## 4.4 ABLATION STUDIES

**The effect of adaptive $\beta$-VAE.**  We assess the effectiveness of scheduling the weighting coefficient $\beta$ in the VAE model. Figure 3 presents the trends of the reconstruction loss and the KL-divergence loss with the scheduled $\beta$ and constant $\beta$ values (from $10^{-1}$ to $10^{-5}$) across 4,000 training epochs. Notably, a large $\beta$ value leads to subpar reconstruction, while a small $\beta$ value results in a large divergence

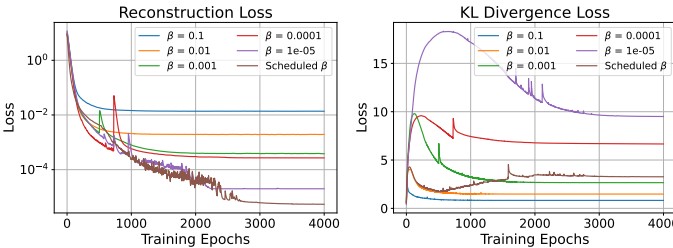

Figure 3: The trends of the validation reconstruction (left) and KL-divergence (right) losses on the Adult dataset, with varying constant $\beta$, and our proposed scheduled $\beta$ ($\beta_{\max} = 0.01, \beta_{\min} = 10^{-5}, \lambda = 0.7$). The proposed scheduled $\beta$ obtains the lowest reconstruction loss with a fairly low KL-divergence loss.

Table 4: The results of single-column density and pair-wise column correlation estimation with different $\beta$ values on the Adult dataset.

| $\beta$ | Single | Pair |
|---|---|---|
| $10^{-1}$ | 1.24 | 3.05 |
| $10^{-2}$ | 0.87 | 2.79 |
| $10^{-3}$ | 0.72 | 2.25 |
| $10^{-4}$ | 0.69 | 2.01 |
| $10^{-5}$ | 41.96 | 69.17 |
| Scheduled $\beta$ | **0.58** | **1.54** |

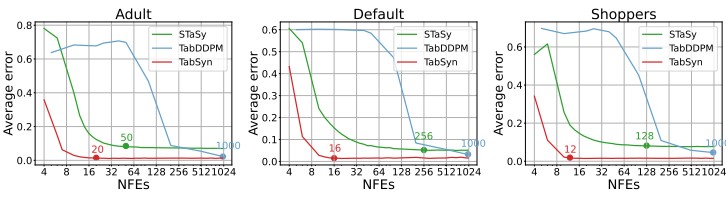

Figure 4: Quality of synthetic data as a function of NFEs on STaSy, TabDDPM, and TABSYN. TABSYN can generate synthetic data of the best quality with fewer NFEs (indicating faster sampling speed).

Table 5: Performance of TABSYN's variants on Adult dataset, on low-order statistics estimation tasks.

| Variants | Single | Pair |
|---|---|---|
| TabDDPM | 1.75 | 3.01 |
| TABSYN-OneHot | 5.59 | 6.92 |
| TABSYN-DDPM | 1.02 | 2.15 |
| TABSYN | **0.58** | **1.54** |

between the embedding distribution and the standard Gaussian, making the balance hard to achieve. In contrast, by dynamically scheduling $\beta$ during training ($\beta_{\max} = 0.01, \beta_{\min} = 10^{-5}, \lambda = 0.7$), we not only prevent excessive KL divergence but also enhance quality. Table 4 further evaluates the learned embeddings from various $\beta$ values of the VAE model via synthetic data quality (single-column density and pair-wise column correlation estimation tasks). This demonstrates the superior performance of our proposed scheduled $\beta$ approach to train the VAE model.

**The effect of linear noise levels.** We evaluate the effectiveness of using linear noise levels, $\sigma(t) = t$, in the diffusion process. As Section 3.3 outlines, linear noises lead to linear trajectories and faster sampling speed. Consequently, we compare TABSYN and two other diffusion models (STaSy and TabDDPM) in terms of the single-column density and pair-wise column correlation estimation errors relative to the number of function evaluations (NFEs), i.e., denoising steps to generate the real data. As continuous-time diffusion models, the proposed TABSYN and STaSy are flexible in choosing NFEs. For TabDDPM, we use the DDIM sampler (Song et al., 2021a) to adjust NFEs. Figure 4 shows that TABSYN not only significantly improves the sampling speed but also consistently yields better performance (with fewer than 20 NFEs for optimal results). In contrast, STaSy requires 50-200 NFEs, varying by datasets, and achieves sub-optimal performance. TabDDPM achieves competitive performance with 1,000 NFEs but significantly drops in performance when reducing NFEs.

**Comparing different encoding/diffusion methods.** We assess the effectiveness of learning the diffusion model in the latent space learned by VAE by creating two TABSYN variants: 1) TABSYN-OneHot: replacing VAE with one-hot encodings of categorical variables and 2) TABSYN-DDPM: substituting the diffusion process in Equation (5) with DDPM as used in TabDDPM. Results in Table 5 demonstrate: 1) One-hot encodings for categorical variables plus continuous diffusion models lead to the worst performance, indicating that it is not appropriate to treat categorical columns simply as continuous features; 2) TABSYN-DDPM in the latent space outperforms TabDDPM in the data space, highlighting the benefit of learning high-quality latent embeddings for improved diffusion modeling; 3) TABSYN surpasses TABSYN-DDPM, indicating the advantage of employing tailored diffusion models in the continuous latent space for better data distribution learning.

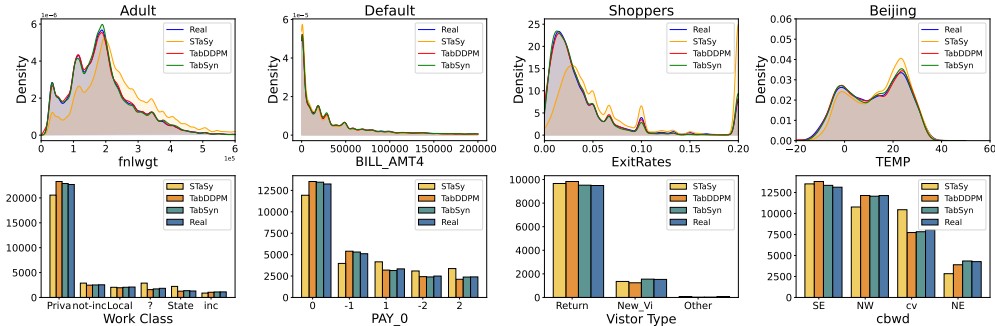

Figure 5: Visualization of synthetic data's single column distribution density (from STaSy, TabDDPM, and TABSYN) v.s. the real data. Upper: numerical columns; Lower: Categorical columns. Note that numerical columns show competitive performance with baselines, while TABSYN excels in estimating categorical column distributions.

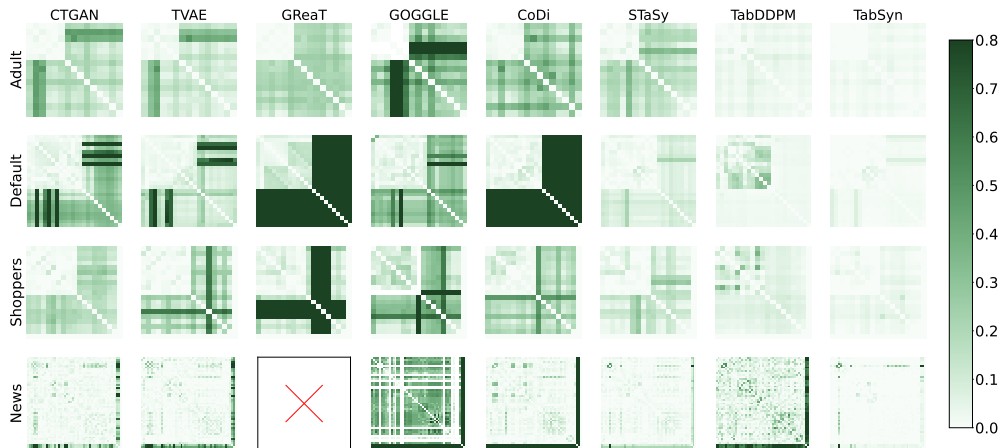

Figure 6: Heatmaps of the pair-wise column correlation of synthetic data v.s. the real data. The value represents the absolute divergence between the real and estimated correlations (the lighter, the better). TABSYN gives the most accurate column correlation estimation.

## 4.5 VISUALIZATION

In Figure 5, we compare column density across eight columns from four datasets (one numerical and one categorical per dataset). TabDDPM matches TABSYN's accuracy on numerical columns but falls short on categorical ones. Figure 6 displays the divergence heatmap between estimated pair-wise column correlations and real correlations. TABSYN gives the most accurate correlation estimation, while other methods exhibit suboptimal performance. These results justify that employing generative modeling in latent space enhances the learning of categorical features and joint column distributions.

## 5 CONCLUSIONS

In this paper, we have proposed TABSYN for synthetic tabular data generation. The TABSYN framework leverages a VAE to map tabular data into a latent space, followed by utilizing a diffusion-based generative model to learn the latent distribution. This approach presents the dual advantages of accommodating numerical and categorical features within a unified latent space, thus facilitating a more comprehensive understanding of their interrelationships and enabling the utilization of advanced generative models in a continuous embedding space. To address potential challenges, TABSYN proposes a model design and training methods, resulting in a highly stable generative model. In addition, TABSYN rectifies the deficiency in prior research by employing a diverse set of evaluation metrics to comprehensively compare the proposed method with existing approaches, showcasing the remarkable quality and fidelity of the generated samples in capturing the original data distribution.

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

# A  ALGORITHMS

In this section, we provide an algorithmic illustration of the proposed TABSYN. Algorithm 1 and Algorithm 2 present the algorithms of the VAE and Diffusion phases of the training process of TABSYN, respectively. Algorithm 3 presents TABSYN's sampling algorithm.

---

**Algorithm 1:** TABSYN: Training of VAE

1: Sample $z = (z^{\mathrm{num}}, z^{\mathrm{cat}}) \sim p(\mathcal{T})$
2: Get tokenized feature $e$ via Eq. 1
3: Get $\mu$ and $\log \sigma$ via VAE's Transformer encoder
4: Reparameterization: $\hat{z} = \mu + \varepsilon \cdot \sigma$, where $\varepsilon \sim \mathcal{N}(\mathbf{0}, \mathbf{I})$
5: Get $\hat{e}$ via VAE's Transformer decoder
6: Get detokenized feature $\hat{z}$ via Eq. 3
7: Calculate loss $\mathcal{L} = \ell_{\mathrm{recon}}(z, \hat{z}) + \beta \ell_{\mathrm{kl}}(\mu, \sigma)$
8: Update the network parameter via Adam optimizer
9: **if** $\ell_{\mathrm{recon}}$ fails to decrease for $S$ steps **then**
10:    $\beta \leftarrow \lambda \beta$
11: **end if**

---

**Algorithm 2:** TABSYN: Training of Diffusion

1: Sample the embedding $z_0$ from $p(z) = p(\mu)$
2: Sample time steps $t$ from $p(t)$ then get $\sigma(t)$
3: Sample noise vectors $\varepsilon \sim \mathcal{N}(\mathbf{0}, \sigma_i^2 \mathbf{I})$
4: Get perturbed data $z_t = z_0 + \varepsilon$
5: Calculate loss $\ell(\theta) = \|\epsilon_\theta(z_t, t) - \varepsilon\|_2^2$
6: Update the network parameter $\theta$ via Adam optimizer

---

**Algorithm 3:** TABSYN: Sampling

1: Sample $z_T \sim \mathcal{N}(\mathbf{0}, \sigma^2(T)\mathbf{I}), t_{\max} = T$
2: **for** $i = \max, \cdots, 1$ **do**
3:    $\nabla_{z_{t_i}} \log p(z_{t_i}) = -\epsilon_\theta(z_{t_i}, t_i)/\sigma(t_i)$
4:    get $z_{t_{i-1}}$ via solving the SDE in Eq. 6.
5: **end for**
6: Put $z_0$ as input of the VAE's Transformer decoder, then acquire $\hat{e}$
7: Get detokenized feature $\hat{z}$ via Eq. 3
8: $\hat{z}$ is the sampled synthetic data

---

# B  DIFFUSION MODELS BASICS

Diffusion models are often presented as a pair of two processes.

- A fixed forward process governs the training of the model, which adds Gaussian noises of increasing scales to the original data.

- A corresponding backward process involves utilizing the trained model iteratively to denoise the samples starting from a fully noisy prior distribution.

## B.1  FORWARD PROCESS

Although there are different mathematical formulations (discrete or continuous) of the diffusion model, Song et al. (2021b) provides a unified formulation via the Stochastic Differential Equation (SDE) and defines the forward process of Diffusion as (note that in this paper, the independent variable is denoted as $z$)

$$\mathrm{d}z = \boldsymbol{f}(z, t)\mathrm{d}t + g(t)\,\mathrm{d}\boldsymbol{w}_t, \tag{8}$$

where $\boldsymbol{f}(\cdot)$ and $g(\cdot)$ are the drift and diffusion coefficients and are selected differently for different diffusion processes, e.g., the variance preserving (VP) and variance exploding (VE) formulations. $\boldsymbol{\omega}_t$ is the standard Wiener process. Usually, $f(\cdot)$ is of the form $\boldsymbol{f}(z, t) = f(t)\, z$. Thus, the SDE can be equivalently written as

$$\mathrm{d}z = f(t)\, z\, \mathrm{d}t + g(t)\,\mathrm{d}\boldsymbol{w}_t. \tag{9}$$

Let $z$ be a function of the time $t$, i.e., $z_t = z(t)$, then the conditional distribution of $z_t$ given $z_0$ (named as the perturbation kernel of the SDE) could be formulated as:

$$p(z_t|z_0) = \mathcal{N}(z_t; s(t)z_0, s^2(t)\sigma^2(t)I), \tag{10}$$

where

$$s(t) = \exp\left(\int_0^t f(\xi)\mathrm{d}\xi\right), \text{ and } \sigma(t) = \sqrt{\int_0^t \frac{g^2(\xi)}{s^2(\xi)}\mathrm{d}\xi}. \tag{11}$$

Therefore, the forward diffusion process could be equivalently formulated by defining the perturbation kernels (via defining appropriate $s(t)$ and $\sigma(t)$).

Variance Preserving (VP) implements the perturbation kernel Eq. 10 by setting $s(t) = \sqrt{1 - \beta(t)}$, and $\sigma(t) = \sqrt{\frac{\beta(t)}{1-\beta(t)}}$ ($s^2(t) + s^2(t)\sigma^2(t) = 1$). Denoising Diffusion Probabilistic Models (DDPM, Ho et al. (2020)) belong to VP-SDE by using discrete finite time steps and giving specific functional definitions of $\beta(t)$.

Variance Exploding (VE) implements the perturbation kernel Eq. 10 by setting $s(t) = 1$, indicating that the noise is directly added to the data rather than weighted mixing. Therefore, The noise variance (the noise level) is totally decided by $\sigma(t)$. The diffusion model used in TABSYN belongs to VE-SDE, but we use linear noise level (i.e., $\sigma(t) = t$) rather than $\sigma(t) = \sqrt{t}$ in the vanilla VE-SDE (Song et al., 2021b). When $s(t) = 1$, the perturbation kernels become:

$$p(z_t|z_0) = \mathcal{N}(z_t; 0, \sigma^2(t)I) \;\Rightarrow\; z_t = z_0 + \sigma(t)\varepsilon, \tag{12}$$

which aligns with the forward diffusion process in Eq. 5.

## B.2 Reverse Process

The sampling process of diffusion models is defined by a corresponding reverse SDE:

$$\mathrm{d}z = [f(z, t) - g^2(t)\nabla_z \log p_t(z)]\mathrm{d}t + g(t)\mathrm{d}w_t. \tag{13}$$

For VE-SDE, $s(t) = 1 \Leftrightarrow f(z, t) = f(t) \cdot z = 0$, and

$$\begin{aligned}
\sigma(t) &= \sqrt{\int_0^t g^2(\xi)\mathrm{d}\xi} \Rightarrow \int_0^t g^2(\xi)\mathrm{d}\xi = \sigma^2(t), \\
g^2(t) &= \frac{\mathrm{d}\sigma^2(t)}{\mathrm{d}t} = 2\sigma(t)\dot{\sigma}(t), \\
g(t) &= \sqrt{2\sigma(t)\dot{\sigma}(t)}.
\end{aligned} \tag{14}$$

Plugging $g(t)$ into Eq. 13, the reverse process in Eq. 6 is recovered:

$$\mathrm{d}z = -2\sigma(t)\dot{\sigma}(t)\nabla_z \log p_t(z)\mathrm{d}t + \sqrt{2\sigma(t)\dot{\sigma}(t)}\mathrm{d}\omega_t. \tag{15}$$

## B.3 Training

As $f(z, t), g(t), w_t$ are all known, if $\nabla_z \log p_t(z)$ (named the score function) is also available, we can sample synthetic data via the reverse process from random noises. Diffusion models train a neural network (named the denoising function) $D_\theta(z_t, t)$ to approximate $\nabla_z \log p_t(z)$. However, $\nabla_z \log p_t(z)$ itself is intractable, as the marginal distribution $p_t(z) = p(z_t)$ is intractable. Fortunately, the conditional distribution $p(z_t|z_0)$ is tractable. Therefore, we can train the denoising function to approximate the conditional score function instead $\nabla_{z_t} \log p(z_t|z_0)$, and the training process is called denoising score matching:

$$\min \mathbb{E}_{z_0 \sim p(z_0)}\mathbb{E}_{z_t \sim p(z_t|z_0)}\|D_\theta(z_t, t) - \nabla_{z_t} \log p(z_t|z_0))\|_2^2, \tag{16}$$

where $\nabla_{\boldsymbol{z}} \log p(\boldsymbol{z}_t | \boldsymbol{z}_0)$ has analytical solution according to Eq. 10:

$$
\begin{aligned}
\nabla_{\boldsymbol{z}_t} \log p(\boldsymbol{z}_t | \boldsymbol{z}_0) &= \frac{1}{p(\boldsymbol{z}_t | \boldsymbol{z}_0)} \nabla_{\boldsymbol{z}_t} p(\boldsymbol{z}_t | \boldsymbol{z}_0) \\
&= \frac{1}{p(\boldsymbol{z}_t | \boldsymbol{z}_0)} \cdot \left( -\frac{1}{s^2(t)\sigma^2(t)} (\boldsymbol{z}_t - s(t)\boldsymbol{z}_0) \right) \cdot p(\boldsymbol{z}_t | \boldsymbol{z}_0) \\
&= -\frac{1}{s^2(t)\sigma^2(t)} (\boldsymbol{z}_t - s(t)\boldsymbol{z}_0) \\
&= -\frac{1}{s^2(t)\sigma^2(t)} \left( s(t)\boldsymbol{z}_0 + s(t)\sigma(t)\boldsymbol{\varepsilon} - s(t)\boldsymbol{z}_0 \right) \\
&= -\frac{\boldsymbol{\varepsilon}}{s(t)\sigma(t)}.
\end{aligned}
\tag{17}
$$

Therefore, Eq. 16 becomes

$$
\begin{aligned}
&\min \mathbb{E}_{\boldsymbol{z}_0 \sim p(\boldsymbol{z}_0)} \mathbb{E}_{\boldsymbol{z}_t \sim p(\boldsymbol{z}_t | \boldsymbol{z}_0)} \| D_\theta(\boldsymbol{z}_t, t) + \frac{\boldsymbol{\varepsilon}}{s(t)\sigma(t)} \|_2^2 \\
&\Rightarrow \min \mathbb{E}_{\boldsymbol{z}_0 \sim p(\boldsymbol{z}_0)} \mathbb{E}_{\boldsymbol{z}_t \sim p(\boldsymbol{z}_t | \boldsymbol{z}_0)} \| \boldsymbol{\epsilon}_\theta(\boldsymbol{z}_t, t) - \boldsymbol{\varepsilon} \|_2^2,
\end{aligned}
\tag{18}
$$

where $D_\theta(\boldsymbol{z}_t, t) = -\frac{\boldsymbol{\epsilon}_\theta(\boldsymbol{z}_t, t)}{s(t)\sigma(t)}$. After the training ends, sampling is enabled by solving Eq. 13 (replacing $\nabla_{\boldsymbol{z}} \log p_t(\boldsymbol{z})$ with $D_\theta(\boldsymbol{z}_t, t)$).

## C PROOFS

### C.1 PROOF FOR PROPOSITION 1

We first introduce Lemma 1 (from (Karras et al., 2022)), which introduces a family of SDEs sharing the same solution trajectories with different forwarding processes:

**Lemma 1.** *Let $g(t)$ be a free parameter functional of $t$, and the following family of (forward) SDEs have the same marginal distributions of the solution trajectories with noise levels $\sigma(t)$ for any choice of $g(t)$:*

$$
\mathrm{d}\boldsymbol{z} = \left( \frac{1}{2} g^2(t) - \dot{\sigma}(t)\sigma(t) \right) \nabla_{\boldsymbol{z}} \log p(\boldsymbol{z}; \sigma(t)) \mathrm{d}t + g(t) \mathrm{d}\omega_t.
\tag{19}
$$

The reverse family of SDEs of Eq. 19 is given by changing the sign of the first term:

$$
\mathrm{d}\boldsymbol{z} = -\frac{1}{2} g^2(t) \nabla_{\boldsymbol{z}} \log p(\boldsymbol{z}; \sigma(t) - \dot{\sigma}(t)\sigma(t) \nabla_{\boldsymbol{z}} \log p(\boldsymbol{z}; \sigma(t)) \mathrm{d}t + g(t) \mathrm{d}\omega_t.
\tag{20}
$$

Lemma 1 indicates that for a specific (forward) SDE following Eq. 19, we can obtain its solution trajectory by solving Eq. 20 of any $g(t)$.

Since our forwarding diffusion process (Eq. 5) lets $g(t) = \sqrt{2\dot{\sigma}(t)\sigma(t)}$ (see derivations in Appendix B), its solution trajectory can be solved via letting $g(t) = 0$ in Eq. 20:

$$
\begin{aligned}
\mathrm{d}\boldsymbol{z} &= -\dot{\sigma}(t)\sigma(t) \nabla_{\boldsymbol{z}} \log p(\boldsymbol{z}; \sigma(t)) \mathrm{d}t, \\
\frac{\mathrm{d}\boldsymbol{z}}{\mathrm{d}t} &= -\dot{\sigma}(t)\sigma(t) \nabla_{\boldsymbol{z}} \log p(\boldsymbol{z}; \sigma(t)).
\end{aligned}
\tag{21}
$$

Eq. 21 is usually named the Probability-Flow ODE, since it depicts a deterministic reverse process without noise terms. Based on Lemma 1, we can study the solution of Eq. 6 using Eq. 21.

To prove Proposition 1, we only have to study the absolute error between the ground-truth $\boldsymbol{z}_{t-\Delta t}$ the approximated one by solving Eq. 6 from $\boldsymbol{z}_t$, where $\Delta t \to 0$:

$$
\| \boldsymbol{z}_{t-\Delta t} - \boldsymbol{z}_{t-\Delta t}^{\text{Euler}} \|.
\tag{22}
$$

Since $\boldsymbol{z}_t = \boldsymbol{z}_0 + \sigma(t)\boldsymbol{\varepsilon}$, there is $\boldsymbol{z}_{t-\Delta t} = \boldsymbol{z}_t - \boldsymbol{\varepsilon} \cdot (\sigma(t) - \sigma(t - \Delta t))$.

The solution of $z_{t-\Delta t}$ from $z_t$ can be obtained using 1st-order Euler method:

$$
\begin{aligned}
z_{t-\Delta t}^{\text{Euler}} &\approx z_t - \Delta t \frac{\mathrm{d}z_t}{\mathrm{d}t} \\
&= z_t + \Delta t \; \dot{\sigma}(t)\sigma(t)\nabla_{z_t}\log p(z_t) \\
&= z_t + \Delta t \; \dot{\sigma}(t)\sigma(t)(-\epsilon_\theta(z_t,t)/\sigma(t)) \\
&= z_t - \dot{\sigma}(t)\epsilon_\theta(z_t,t)\Delta t \\
&= z_t - \dot{\sigma}(t)\varepsilon\Delta t,
\end{aligned}
\tag{23}
$$

$$
\begin{aligned}
\|z_{t-\Delta t} - z_{t-\Delta t}^{\text{Euler}}\| &= \varepsilon \cdot (\sigma(t) - \sigma(t-\Delta t)) - \dot{\sigma}(t)\varepsilon\Delta t \\
&= \varepsilon \cdot (\sigma(t) - \sigma(t-\Delta t) - \dot{\sigma}(t)\Delta t).
\end{aligned}
\tag{24}
$$

Specifically, if $\sigma(t) = t, \dot{\sigma}(t) = 1$, there is

$$
\|z_{t-\Delta t} - z_{t-\Delta t}^{\text{Euler}}\| = \mathbf{0}.
\tag{25}
$$

Comparably, setting $\sigma(t) = \sqrt{t}$ (as in VE-SDE Song et al. (2021b)) leads to

$$
\|z_{t-\Delta t} - z_{t-\Delta t}^{\text{Euler}}\| = |\epsilon(\sqrt{t} - \sqrt{t-\Delta t} - \frac{\Delta t}{2\sqrt{t}})| \geq \mathbf{0}.
\tag{26}
$$

Therefore, the proof is complete.

## D   NETWORK ARCHITECTURES

### D.1   ARCHITECTURES OF VAE

In Sec. 3.2, we introduce the general architecture of our VAE module, which consists of a tokenizer, a Transformer encoder, a Transformer decoder, and a detokenizer. Figure 7 is a detailed illustration of the architectures of the Transformer encoder and decoder.

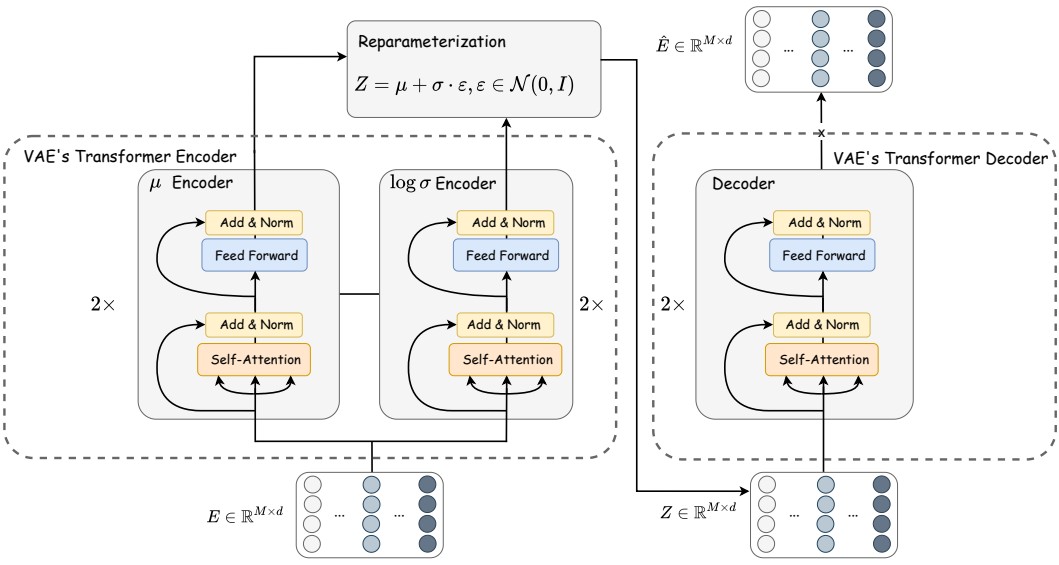

Figure 7: Architectures of VAE's Encoder and Decoder. Both the encoder and decoder are implemented as a two-layer Transformer of identical architectures.

The VAE's encoder takes the tokenizer's output $\boldsymbol{E} \in \mathbb{R}^{M \times d}$ as input. As we are using the Variational AutoEncoder, the encoder module consists of a $\mu$ encoder and a $\log \sigma$ encoder of the same architecture.

Each encoder is implemented as a two-layer Transformer, each with a Self-Attention (without multi-head) module and a Feed Forward Neural Network (FFNN). The FFNN used in TABSYN is a simple two-layer MLP with ReLU activation(the input of the FFNN is denoted by $\boldsymbol{H}_0$):

$$
\begin{aligned}
\boldsymbol{H}_1 &= \texttt{ReLU}(\texttt{FC}(\boldsymbol{H}_0)) \in \mathbb{R}^{M \times D}, \\
\boldsymbol{H}_2 &= \texttt{FC}(\boldsymbol{H}_1), \in \mathbb{R}^{M \times d},
\end{aligned}
\tag{27}
$$

where $\texttt{FC}$ denotes the fully-connected layer, and $D$ is FFNN's hidden dimension. In this paper, we set $d = 4$ and $D = 128$ for all datasets. "Add & Norm" in Figure 7 denotes residual connection and Layer Normalization (Ba et al., 2016), respectively.

The VAE encoder outputs two matrixes: mean matrix $\boldsymbol{\mu} \in \mathbb{R}^{M \times d}$ and log standard deviation matrix $\log \boldsymbol{\sigma} \in \mathbb{R}^{M \times d}$. Then, the latent variables are obtained via the parameterization trick:

$$
\boldsymbol{Z} = \boldsymbol{\mu} + \boldsymbol{\sigma} \cdot \boldsymbol{\varepsilon}, \boldsymbol{\varepsilon} \sim \mathcal{N}(\boldsymbol{0}, \boldsymbol{I}).
\tag{28}
$$

The VAE's decoder is another two-layer Transformer of the same architecture as the encoder, and it takes $\boldsymbol{Z}$ as input. The decoder is expected to output $\hat{\boldsymbol{E}} \in \mathbb{R}^{M \times d}$ for the detokenizer.

## D.2 ARCHITECTURES OF DENOISING MLP

In Figure 7, we present the architecture of the denoising neural networks $\boldsymbol{\varepsilon}_\theta(\boldsymbol{z}_t, t)$ in Eq. 7, which is a simple MLP of the similar architecture as used in TabDDPM (Kotelnikov et al., 2023).

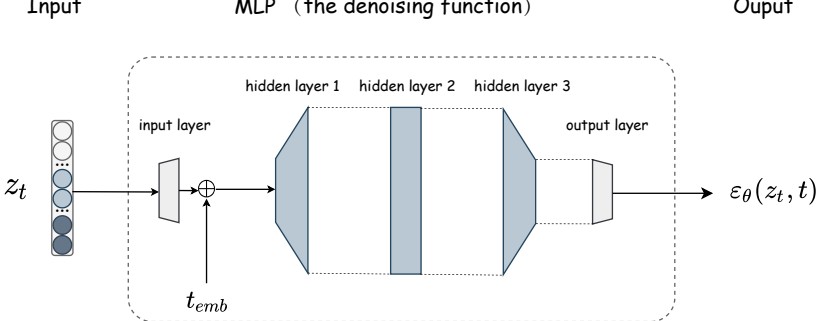

Figure 8: Architectures of denoising function $\boldsymbol{\varepsilon}_\theta(\boldsymbol{z}_t, t)$. The denoising function is a 5-layer MLP with SiLU activations. $t_{\text{emb}}$ is the sinusoidal timestep embeddings.

The denoising MLP takes the current time step $t$ and the corresponding latent vector $\boldsymbol{z}_t \in \mathbb{R}^{1 \times Md}$ as input. First, $\boldsymbol{z}_t$ is fed into a linear projection layer that converts the vector dimension to be $d_{\text{hidden}}$:

$$
\boldsymbol{h}_0 = \texttt{FC}_{\text{in}}(\boldsymbol{z}_t) \in \mathbb{R}^{1 \times d_{\text{hidden}}},
\tag{29}
$$

where $\boldsymbol{h}_0$ is the transformed vector, and $d_{\text{hidden}}$ is the output dimension of the input layer.

Then, following the practice in TabDDPM (Kotelnikov et al., 2023), the sinusoidal timestep embeddings $\boldsymbol{t}_{\text{emb}} \in \mathbb{R}^{1 \times d_{\text{hidden}}}$ is added to $\boldsymbol{h}_0$ to obtain the input vector $\boldsymbol{h}_{\text{hidden}}$:

$$
\boldsymbol{h}_{\text{in}} = \boldsymbol{h}_0 + \boldsymbol{t}_{\text{emb}}.
\tag{30}
$$

The hidden layers are three fully connected layers of the size $d_{\text{hidden}} - 2 * d_{\text{hidden}} - 2 * d_{\text{hidden}} - d_{\text{hidden}}$, with SiLU activation functions (in consistency with TabDDPM (Kotelnikov et al., 2023)):

$$
\begin{aligned}
\boldsymbol{h}_1 &= \texttt{SiLU}(\texttt{FC}_1(\boldsymbol{h}_0) \in \mathbb{R}^{1 \times 2 * d_{\text{hidden}}}), \\
\boldsymbol{h}_2 &= \texttt{SiLU}(\texttt{FC}_2(\boldsymbol{h}_1) \in \mathbb{R}^{1 \times 2 * d_{\text{hidden}}}), \\
\boldsymbol{h}_3 &= \texttt{SiLU}(\texttt{FC}_3(\boldsymbol{h}_2) \in \mathbb{R}^{1 \times d_{\text{hidden}}}).
\end{aligned}
\tag{31}
$$

The estimated score is obtained via the last linear layer:

$$
\boldsymbol{\varepsilon}_\theta(\boldsymbol{z}_t, t) = \boldsymbol{h}_{\text{out}} = \texttt{FC}_{\text{out}}(\boldsymbol{h}_3) \in \mathbb{R}^{1 \times d_{\text{in}}}.
\tag{32}
$$

Finally, $\boldsymbol{\varepsilon}_\theta(\boldsymbol{z}_t, t)$ is applied to Eq. 7 for model training.

# E    DETAILS OF EXPERIMENTAL SETUPS

We implement TABSYN and all the baseline methods with PyTorch. All the methods are optimized with Adam (Kingma & Ba, 2015) optimizer. All the experiments are conducted on an Nvidia RTX 4090 GPU with 24G memory.

## E.1    DATASETS

We use 6 tabular datasets from UCI Machine Learning Repository[1]: Adult, Default, Shoppers, Magic, Beijing, and News, where each tabular dataset is associated with a machine-learning task. Classification: Adult, Default, Magic, and Shoppers. Regression: Beijing and News. The statistics of the datasets are presented in Table 6.

Table 6: Statistics of datasets. # Num stands for the number of numerical columns, and # Cat stands for the number of categorical columns.

| Dataset | # Rows | # Num | # Cat | # Train | # Validation | # Test | Task |
|---------|--------|-------|-------|---------|--------------|--------|------|
| **Adult** | $48,842$ | 6 | 9 | $28,943$ | $3,618$ | $16,281$ | Classification |
| **Default** | $30,000$ | 14 | 11 | $24,000$ | $3,000$ | $3,000$ | Classification |
| **Shoppers** | $12,330$ | 10 | 8 | $9,864$ | $1,233$ | $1,233$ | Classification |
| **Magic** | $19,019$ | 10 | 1 | $15,215$ | $1,902$ | $1,902$ | Classification |
| **Beijing** | $43,824$ | 7 | 5 | $35,058$ | $4,383$ | $4,383$ | Regression |
| **News** | $39,644$ | 46 | 2 | $31,714$ | $3,965$ | $3,965$ | Regression |

In Table 6, # Rows denote the number of rows (records) in the table. # Num and # Cat denote the number of numerical features and categorical features, respectively. Note that the target column is counted as either a numerical or a categorical feature, depending on the task type. Specifically, the target column belongs to the categorical column if the task is classification; otherwise, it is a numerical column. Each dataset is split into training, validation, and testing sets for the Machine Learning Efficiency experiments. As Adult has its official testing set, we directly use it as the testing set. The original training set of Adult is further split into training and validation split with the ratio $8 : 1$. The remaining datasets are split into training/validation/testing sets with the ratio $8 : 1 : 1$ with a fixed seed.

Below is a detailed introduction to each dataset:

- **Adult**[2]: The "**Adult** Census Income" dataset contains the demographic and employment-related features people. The task is to predict whether an individual's income exceeds $50,000$.

- **Default**[3]: The "**Default** of Credit Card Clients Dataset" dataset contains information on default payments, demographic factors, credit data, history of payment, and bill statements of credit card clients in Taiwan from April 2005 to September 2005. The task is to predict whether the client will default payment next month.

- **Shoppers**[4]: The "Online **Shoppers** Purchasing Intention Dataset" contains information of user's webpage visiting sessions. The task is to predict if the user's session ends with the shopping behavior.

- **Magic**[5]: The "**Magic** Gamma Telescope" dataset is to simulate registration of high energy gamma particles in a ground-based atmospheric Cherenkov gamma telescope using the imaging technique. The task is to classify high-energy Gamma particles in the atmosphere.

---

[1]https://archive.ics.uci.edu/datasets
[2]https://archive.ics.uci.edu/dataset/2/adult
[3]https://archive.ics.uci.edu/dataset/350/default+of+credit+card+clients
[4]https://archive.ics.uci.edu/dataset/468/online+shoppers+purchasing+intention+dataset
[5]https://archive.ics.uci.edu/dataset/159/magic+gamma+telescope

- **Beijing**[6]: The "**Beijing** PM2.5 Data" dataset contains the hourly PM2.5 data of US Embassy in Beijing and the meteorological data from Beijing Capital International Airport. The task is to predict the PM2.5 value.
- **News**[7]: The "Online **News** Popularity" dataset contains a heterogeneous set of features about articles published by Mashable in two years. The goal is to predict the number of shares in social networks (popularity).

### E.2 BASELINES

In this section, we introduce and compare the properties of the baseline methods used in this paper.

- **CTGAN** and **TVAE** are two methods for synthetic tabular data generation proposed in one paper (Xu et al., 2019), using the same techniques proposed but based on different basic generative models – **GAN** for CTGAN while **VAE** for TVAE. The two methods contain two important designs: 1) Mode-specific Normalization to deal with numerical columns with complicated distributions. 2) Conditional Generation of numerical columns based on categorical columns to deal with class imbalance problems.

- **GOGGLE** (Liu et al., 2023b) is a recently proposed synthetic tabular data generation model based on **VAE**. The primary motivation of GOGGLE is that the complicated relationships between different columns are hardly exploited by previous literature. Therefore, it proposes to learn a graph adjacency matrix to model the dependency relationships between different columns. The encoder and decoder of the VAE model are both implemented as Graph Neural Networks (GNNs), and the graph adjacent matrix is jointly optimized with the GNNs parameters.

- **GReaT** (Borisov et al., 2023) treats a row of tabular data as a sentence and applies the Auto-regressive GPT model to learn the sentence-level row distributions. GReaT involves a well-designed serialization process to transform a row into a natural language sentence of a specific format and a corresponding deserialization process to transform a sentence back to the table format. To ensure the permutation invariant property of tabular data, GReaT shuffles each row several times before serialization.

- **STaSy** (Kim et al., 2023) is a recent diffusion-based model for synthetic tabular data generation. STaSy treats the one-hot encoding of categorical columns as continuous features, which are then processed together with the numerical columns. STaSy adopts the VP/VE SDEs from Song et al. (2021b) as the diffusion process to learn the distribution of tabular data. STaSy further proposes several training strategies, including self-paced learning and fine-tuning, to stabilize the training process, increasing sampling quality and diversity.

- **CoDi** (Lee et al., 2023) proposes to utilize two diffusion models for numerical and categorical columns, respectively. For numerical columns, it uses the DDPM (Ho et al., 2020) model with Gaussian noises. For categorical columns, it uses the multinominal diffusion model (Hoogeboom et al., 2021) with categorical noises. The two diffusion processes are inter-conditioned on each other to model the joint distribution of numerical and categorical columns. In addition, CoDi adopts contrastive learning methods to further bind the two diffusion methods.

- **TabDDPM** (Kotelnikov et al., 2023). Like CoDi, TabDDPM introduces two diffusion processes: DDPM with Gaussian noises for numerical columns and multinominal diffusion with categorical noises for categorical columns. Unlike CoDi, which introduces many additional techniques, such as co-evolved learning via inter-conditioning and contrastive learning, TabDDPM concatenates the numerical and categorical features as input and output of the denoising function (an MLP). Despite its simplicity, our experiments have shown that TabDDPM performs even better than CoDi.

We further compare the properties of these baseline methods and the proposed TABSYN in Table 7. The compared properties include: 1) **Compatibility**: if the method can deal with mixed-type data columns, e.g., numerical and categorical. 2) **Robustness**: if the method has stable performance across

---

[6] https://archive.ics.uci.edu/dataset/381/beijing+pm2+5+data
[7] https://archive.ics.uci.edu/dataset/332/online+news+popularity

different datasets (measured by the standard deviation of the scores ($\leq 10\%$ or not) on different datasets (from Table 1 and Table 2). 3) **Quality**: Whether the synthetic data can pass column-wise Chi-Squared Test ($p \geq 0.95$). 4) **Efficiency**: Each method can generate synthetic tabular data of satisfying quality within less than 20 steps.

Table 7: A comparison of the properties of different generative models for tabular data. *Base model* denotes the base generative model type: Generative Adversarial Networks (GAN), Variational AutoEncoders (VAE), Auto-Regressive Language Models (AR), and Diffusion.

| Method | Base Model | Compatibility | Robustness | Quality | Efficiency |
|--------|-----------|---------------|------------|---------|------------|
| CTGAN | GAN | ✓ | ✗ | ✗ | ✓ |
| TVAE | VAE | ✓ | ✓ | ✗ | ✓ |
| GOGGLE | VAE | ✗ | ✗ | ✗ | ✓ |
| GReaT | AR | ✓ | ✗ | ✗ | ✗ |
| STaSy | Diffusion | ✗ | ✓ | ✓ | ✗ |
| CoDi | Diffusion | ✓ | ✗ | ✗ | ✗ |
| TabDDPM | Diffusion | ✓ | ✗ | ✓ | ✗ |
| TABSYN | Diffusion | ✓ | ✓ | ✓ | ✓ |

### E.3 METRICS OF LOW-ORDER STATISTICS

In this section, we give a detailed introduction of the metrics used in Sec. 4.2.

#### E.3.1 COLUMN-WISE DENSITY ESTIMATION

*Kolmogorov-Sirnov Test (KST)*: Given two (continuous) distributions $p_r(x)$ and $p_s(x)$ ($r$ denotes real and $s$ denotes synthetic), KST quantifies the distance between the two distributions using the upper bound of the discrepancy between two corresponding Cumulative Distribution Functions (CDFs):

$$\text{KST} = \sup_x |F_r(x) - F_s(x)|, \tag{33}$$

where $F_r(x)$ and $F_s(x)$ are the CDFs of $p_r(x)$ and $p_s(x)$, respectively:

$$F(x) = \int_{-\infty}^{x} p(x)\mathrm{d}x. \tag{34}$$

*Total Variation Distance (TVD)*: TVD computes the frequency of each category value and expresses it as a probability. Then, the TVD score is the average difference between the probabilities of the categories:

$$\text{TVD} = \frac{1}{2} \sum_{\omega \in \Omega} |R(\omega) - S(\omega)|, \tag{35}$$

where $\omega$ describes all possible categories in a column $\Omega$. $R(\cdot)$ and $S(\cdot)$ denotes the real and synthetic frequencies of these categories.

#### E.3.2 PAIR-WISE COLUMN CORRELATION

*Pearson Correlation Coefficient*: The Pearson correlation coefficient measures whether two continuous distributions are linearly correlated and is computed as:

$$\rho_{x,y} = \frac{\text{Cov}(x, y)}{\sigma_x \sigma_y}, \tag{36}$$

where $x$ and $y$ are two continuous columns. Cov is the covariance, and $\sigma$ is the standard deviation.

Then, the performance of correlation estimation is measured by the average differences between the real data's correlations and the synthetic data's corrections:

$$\text{Pearson Score} = \frac{1}{2}\mathbb{E}_{x,y}|\rho^R(x, y) - \rho^S(x, y)|, \tag{37}$$

where $\rho^R(x, y)$ and $\rho^S(x, y))$ denotes the Pearson correlation coefficient between column $x$ and column $y$ of the real data and synthetic data, respectively. As $\rho \in [-1, 1]$, the average score is divided by 2 to ensure that it falls in the range of $[0, 1]$, then the smaller the score, the better the estimation.

*Contingency similarity*: For a pair of categorical columns $A$ and $B$, the contingency similarity score computes the difference between the contingency tables using the Total Variation Distance. The process is summarized by the formula below:

$$\text{Contingency Score} = \frac{1}{2} \sum_{\alpha \in A} \sum_{\beta \in B} |R_{\alpha,\beta} - S_{\alpha,\beta}|, \tag{38}$$

where $\alpha$ and $\beta$ describe all the possible categories in column $A$ and column $B$, respectively. $R_{\alpha,\beta}$ and $S_{\alpha,\beta}$ are the joint frequency of $\alpha$ and $\beta$ in the real data and synthetic data, respectively.

### E.4 DETAILS OF MACHINE LEARNING EFFICIENCY EXPERIMENTS

As preliminarily illustrated in Sec. 4.3 and Appendix E.1, each dataset is first split into the real training and testing set. The generative models are learned on the real training set. After the models are learned, a synthetic set of equivalent size is sampled.

The performance of synthetic data on MLE tasks is evaluated based on the divergence of test scores using the real and synthetic training data. Therefore, we first train the machine learning model on the real training set, split into training and validation sets with a $8 : 1$ ratio. The classifier/regressor is trained on the training set, and the optimal hyperparameter setting is selected according to the performance on the validation set. After the optimal hyperparameter setting is obtained, the corresponding classifier/regressor is retrained on the training set and evaluated on the real testing set. We create 20 random splits for training and validation sets, and the performance reported in Table 3 is the mean and std of the AUC/RMSE score over the 20 random trails. The performance of synthetic data is obtained in the same way.

Below is the hyperparameter search space of the XGBoost Classifier/Regressor used in MLE tasks, and we select the optimal hyperparameters via grid search.

- Number of estimators: [10, 50, 100]
- Minimum child weight: [5, 10, 20].
- Maximum tree depth: [1,10].
- Gamma: [0.0, 1.0].

We use the implementations of these metrics from SDMetric[8].

## F ADDITION EXPERIMENTAL RESULTS

In this section, we compare the training and sampling time of different methods, taking the Adult dataset as an example.

### F.1 TRAINING / SAMPLING TIME

As shown in Fig. 8, though having an additional VAE training process, the proposed TABSYN has a similar training time to most of the baseline methods. In regard to the sampling time, TABSYN requires merely 1.784s to generate synthetic data of the same size as Adult's training data, which is close to the one-step sampling methods CTGAN and TVAE. Other diffusion-based methods take a much longer time for sampling. E.g., the most competitive method TabDDPM (Kotelnikov et al., 2023) takes 28.92s for sampling. The proposed TABSYN reduces the sampling time by 93%, and achieves even better synthesis quality.

---

[8]https://docs.sdv.dev/sdmetrics

Table 8: Comparison of training and sampling time of different methods, on Adult dataset. TABSYN's training time is the summation of VAE's and Diffusion's training time.

| Method | Training | Sampling |
|--------|----------|----------|
| CTGAN | 1029.8s | 0.8621s |
| TVAE | 352.6s | 0.5118s |
| GOGGLE | 1h 34min | 5.342s |
| GReaT | 2h 27min | 2min 19s |
| STaSy | 2283s | 8.941s |
| CoDi | 2h 56min | 4.616s |
| TabDDPM | 1031s | 28.92s |
| TABSYN | 1758s + 664s | 1.784s |

## F.2 SAMPLE-WISE QUALITY SCORE OF SYNTHETIC DATA ($\alpha$-PRECISON AND $\beta$-RECALL)

Experiments in Sec. 4 have evaluated the performance of synthetic data generated from different models using low-order statistics, including the column-wise density estimation (Table 1) and pair-wise column correlation estimation (Table 2). However, these results are insufficient to evaluate synthetic data's overall density estimation performance, as the generative model may simply learn to estimate the density of each single column individually rather than the joint probability of all columns. Furthermore, the MLE tasks also cannot reflect the overall density estimation performance since some unimportant columns might be overlooked. Therefore, in this section, we adopt higher-order metrics that focus more on the entire data distribution, i.e., the joint distribution of all the columns.

Following Liu et al. (2023b) and Alaa et al. (2022), we adopt the $\alpha$-Precision and $\beta$-Recall proposed in Alaa et al. (2022), two sample-level metric quantifying how faithful the synthetic data is. In general, $\alpha$-Precision evaluates the fidelity of synthetic data – whether each synthetic example comes from the real-data distribution, $\beta$-Recall evaluates the coverage of the synthetic data, e.g., whether the synthetic data can cover the entire distribution of the real data (In other words, whether a real data sample is close to the synthetic data). Liu et al. (2023b) also adopts the third metric, Authenticity – whether the synthetic sample is generated randomly or simply copied from the real data. However, we found that authenticity score and beta-recall exhibit a predominantly negative correlation – their sum is nearly constant, and an improvement in beta-recall is typically accompanied by a decrease in authenticity score (we believe this is the reason for the relatively small differences in quality scores among the various models in GOGGLE (Liu et al., 2023b)). Therefore, we believe that beta-recall and authenticity are not suitable for simultaneous use.

In Table 9 and Table 10 we report the scores of $\alpha$-Precision and $\beta$-Recall, respectively. TABSYN obtains the best $\alpha$-Precision scores on all the datasets, demonstrating the superior capacity of TABSYN in generating synthetic data that is close to the real ones. In Table 10, we observe that TabSyn consistently achieves high $\beta$-Recall scores across six datasets. Although some baseline methods obtain higher $\beta$-recall scores on specific datasets, it can hardly be concluded that the synthetic data generated by these methods are of better quality since 1) their synthetic data has poor $\alpha$-Precision scores (e.g., GReaT on Adult, and STaSy on Magic), indicating that the synthetic data is far from the real data's manifold; 2) they fail to demonstrate stably competitive performance on other datasets (e.g., GReaT has high $\beta$-Recall scores on Adult but poor scores on Magic). We believe that to assess the quality of generation, the first consideration is whether the generated data is sufficiently authentic ($\alpha$-Precision), and the second is whether the generated data can cover all the modes of the real dataset ($\beta$-Recall). According to this criterion, the quality of data generated by TabSyn is the highest. It not only possesses the highest fidelity score but also consistently demonstrates remarkably high coverage on every dataset.

## F.3 DETECTION: CLASSIFIER TWO SAMPLE TESTS (C2ST)

We further study how difficult it is to tell apart the real data from the synthetic data, therefore evaluating if the synthetic data can recover the real data distribution. To this end, we employ the

Table 9: Comparison of $\alpha$-Precision scores. **Bold Face** represents the best score on each dataset. Higher values indicate superior results. TABSYN outperforms all other baseline methods on all datasets.

| Methods | Adult | Default | Shoppers | Magic | Beijing | News | Average | Ranking |
|---|---|---|---|---|---|---|---|---|
| CTGAN | $77.74_{\pm0.15}$ | $62.08_{\pm0.08}$ | $76.97_{\pm0.39}$ | $86.90_{\pm0.22}$ | $96.27_{\pm0.14}$ | $96.96_{\pm0.17}$ | 82.82 | 5 |
| TVAE | $98.17_{\pm0.17}$ | $85.57_{\pm0.34}$ | $58.19_{\pm0.26}$ | $86.19_{\pm0.48}$ | $97.20_{\pm0.10}$ | $86.41_{\pm0.17}$ | 85.29 | 4 |
| GOGGLE | 50.68 | 68.89 | 86.95 | 90.88 | 88.81 | 86.41 | 78.77 | 8 |
| GReaT | $55.79_{\pm0.03}$ | $85.90_{\pm0.17}$ | $78.88_{\pm0.13}$ | $85.46_{\pm0.54}$ | $98.32_{\pm0.22}$ | - | 80.87 | 6 |
| STaSy | $82.87_{\pm0.26}$ | $90.48_{\pm0.11}$ | $89.65_{\pm0.25}$ | $86.56_{\pm0.19}$ | $89.16_{\pm0.12}$ | $94.76_{\pm0.33}$ | 88.91 | 2 |
| CoDi | $77.58_{\pm0.45}$ | $82.38_{\pm0.15}$ | $94.95_{\pm0.35}$ | $85.01_{\pm0.36}$ | $98.13_{\pm0.38}$ | $87.15_{\pm0.12}$ | 87.03 | 3 |
| TabDDPM | $96.36_{\pm0.20}$ | $97.59_{\pm0.36}$ | $88.55_{\pm0.68}$ | $98.59_{\pm0.17}$ | $97.93_{\pm0.30}$ | $0.00_{\pm0.00}$ | 79.83 | 7 |
| TABSYN | $\mathbf{99.52_{\pm0.10}}$ | $\mathbf{99.26_{\pm0.27}}$ | $\mathbf{99.16_{\pm0.22}}$ | $\mathbf{99.38_{\pm0.27}}$ | $\mathbf{98.47_{\pm0.10}}$ | $\mathbf{96.80_{\pm0.25}}$ | $\mathbf{98.67}$ | 1 |

Table 10: Comparison of $\beta$-Recall scores. **Bold Face** represents the best score on each dataset. Higher values indicate superior results. TABSYN gives consistently high $\beta$-recall scores, indicating that the synthetic data covers a wide range of the real distribution. Although some baseline methods obtain higher scores on specific datasets, they fail to demonstrate stably competitive performance on other datasets.

| Methods | Adult | Default | Shoppers | Magic | Beijing | News | Average | Ranking |
|---|---|---|---|---|---|---|---|---|
| CTGAN | $30.80_{\pm0.20}$ | $18.22_{\pm0.17}$ | $31.80_{\pm0.350}$ | $11.75_{\pm0.20}$ | $34.80_{\pm0.10}$ | $24.97_{\pm0.29}$ | 25.39 | 7 |
| TVAE | $38.87_{\pm0.31}$ | $23.13_{\pm0.11}$ | $19.78_{\pm0.10}$ | $32.44_{\pm0.35}$ | $28.45_{\pm0.08}$ | $29.66_{\pm0.21}$ | 28.72 | 6 |
| GOGGLE | 8.80 | 14.38 | 9.79 | 9.88 | 19.87 | 2.03 | 10.79 | 8 |
| GReaT | $\mathbf{49.12_{\pm0.18}}$ | $42.04_{\pm0.19}$ | $44.90_{\pm0.17}$ | $34.91_{\pm0.28}$ | $43.34_{\pm0.31}$ | - | 43.34 | 2 |
| STaSy | $29.21_{\pm0.34}$ | $39.31_{\pm0.39}$ | $37.24_{\pm0.45}$ | $\mathbf{53.97_{\pm0.57}}$ | $54.79_{\pm0.18}$ | $39.42_{\pm0.32}$ | 42.32 | 3 |
| CoDi | $9.20_{\pm0.15}$ | $19.94_{\pm0.22}$ | $20.82_{\pm0.23}$ | $50.56_{\pm0.31}$ | $52.19_{\pm0.12}$ | $34.40_{\pm0.31}$ | 31.19 | 5 |
| TabDDPM | $47.05_{\pm0.25}$ | $47.83_{\pm0.35}$ | $47.79_{\pm0.25}$ | $48.46_{\pm0.42}$ | $\mathbf{56.92_{\pm0.13}}$ | $0.00_{\pm0.00}$ | 41.34 | 4 |
| TABSYN | $47.56_{\pm0.22}$ | $\mathbf{48.00_{\pm0.35}}$ | $\mathbf{48.95_{\pm0.28}}$ | $48.03_{\pm0.23}$ | $55.84_{\pm0.19}$ | $\mathbf{45.04_{\pm0.34}}$ | $\mathbf{48.90}$ | 1 |

detection metric provided by sdmetrics [9]. In Table 11, we present the results obtained using logistic regression as the detection method. As indicated in the table, the Detection score exhibits superior

Table 11: Detection score (C2ST) using logistic regression classifier. Higher scores indicate better performance.

| Method | Adult | Default | Shoppers | Magic | Beijing | News |
|---|---|---|---|---|---|---|
| SMOTE | 0.9710 | 0.9274 | 0.9086 | 0.9961 | 0.9888 | 0.9344 |
| CTGAN | 0.5949 | 0.4875 | 0.7488 | 0.6728 | 0.7531 | 0.6947 |
| TVAE | 0.6315 | 0.6547 | 0.2962 | 0.7706 | 0.8659 | 0.4076 |
| GOGGLE | 0.1114 | 0.5163 | 0.1418 | 0.9526 | 0.4779 | 0.0745 |
| GReaT | 0.5376 | 0.4710 | 0.4285 | 0.4326 | 0.6893 | - |
| STaSy | 0.4054 | 0.6814 | 0.5482 | 0.6939 | 0.7922 | 0.5287 |
| CoDi | 0.2077 | 0.4595 | 0.2784 | 0.7206 | 0.7177 | 0.0201 |
| TabDDPM | 0.9755 | 0.9712 | 0.8349 | **0.9998** | 0.9513 | 0.0002 |
| TABSYN | **0.9986** | **0.9870** | **0.9740** | 0.9732 | **0.9603** | **0.9749** |

discriminative power compared to other metrics, such as single-column density estimation, pair-wise column shape estimation, and MLE. The detection score shows significant variations across different models for synthetic data generation. As indicated in the table, the Detection score exhibits superior discriminative power compared to other metrics, such as single-column density estimation, pair-wise column shape estimation, and MLE. The detection score shows significant variations across different models for synthetic data generation. The proposed TABSYN consistently achieves notably high scores across all datasets (SMOTE (Chawla et al., 2002) directly interpolates within the training set, so it is not surprising that it achieves high scores in the detection metric.).

---

[9]https://docs.sdv.dev/sdmetrics/metrics/metrics-in-beta/detection-single-table

## F.4 MISSING VALUE IMPUTATION

**Adapting** TABSYN **for missing value imputation** An essential advantage of the Diffusion Model is that an unconditional model can be directly used for missing data imputation tasks (e.g., image inpainting (Song et al., 2021b; Lugmayr et al., 2022) and missing value imputation) without retraining. Following the inpainting methods proposed in Lugmayr et al. (2022), we apply the proposed TABSYN in Missing Value Imputation Tasks.

For a row of tabular data $\boldsymbol{z}_i = [\boldsymbol{z}_i^{\text{num}}, \boldsymbol{z}_{i,1}^{\text{oh}}, \cdots \boldsymbol{z}_{i,M_{\text{cat}}}^{\text{oh}}], \boldsymbol{z}_i^{\text{num}} \in \mathbb{R}^{1 \times M_{\text{num}}}, \boldsymbol{z}_{i,j}^{\text{oh}} \in \mathbb{R}^{1 \times C_j}$. Assume the index set of masked numerical columns is $m_{\text{num}}$, and of masked categorical columns is $m_{\text{cat}}$, we first preprocess the masked columns as follows:

- The value of a masked numerical column is set as the averaged value of this column, i.e., $x_{i,j}^{\text{num}} \Leftarrow \text{mean}(\boldsymbol{z}_{:,j}^{\text{num}}), \forall j \in m_{\text{num}}$.

- The masked categorical column is set as $\boldsymbol{z}_{i,j}^{\text{oh}} \Leftarrow [\frac{1}{C_j}, \cdots, \frac{1}{C_j}, \cdots \frac{1}{C_j}] \in \mathbb{R}^{1 \times C_j}, \forall j \in m_{\text{cat}}$.

The updated $\boldsymbol{z}_i$ (we omit the subscript in the remaining parts) is fed to TABSYN's frozen VAE's encoder to obtain the embedding $\boldsymbol{z} \in \mathbb{R}^{1 \times Md}$. As TABSYN's VAE employs the Transformer architecture, there is a deterministic mapping from the masked indexes in the data space $m_{\text{num}}$ and $m_{\text{cat}}$ to the masked indexes in the latent space. For example, the first dimension of the numerical column is mapped to dimension 1 to $d$ in $\boldsymbol{z}$. Therefore, we can create a masking vector $\boldsymbol{m}$ indicating whether each dimension is masked. Then, the known and unknown part of $\boldsymbol{z}$ could be expressed as $\boldsymbol{m} \odot \boldsymbol{z}$ and $(1 - \boldsymbol{m}) \odot \boldsymbol{z}$, respectively.

Following Lugmayr et al. (2022), the reverse step is modified as a mixture of the known part's forwarding and the unknown part's denoising:

$$
\begin{aligned}
\boldsymbol{z}_{t_{i-1}}^{\text{known}} &= \boldsymbol{z} + \sigma(t_{i-1})\boldsymbol{\varepsilon}, \boldsymbol{\varepsilon} \sim \mathcal{N}(\boldsymbol{0}, \boldsymbol{I}), \\
\boldsymbol{z}_{t_{i-1}}^{\text{unknown}} &= \boldsymbol{z}_{t_i} + \int_{t_i}^{t_{i-1}} \mathrm{d}\boldsymbol{z}_{t_i}, \\
\boldsymbol{z}_{t_{i-1}} &= \boldsymbol{m} \odot \boldsymbol{z}_{t_{i-1}}^{\text{known}} + (1 - \boldsymbol{m}) \odot \boldsymbol{z}_{t_{i-1}}^{\text{unknown}}, \\
\text{where} \quad \mathrm{d}\boldsymbol{z}_t &= -\dot{\sigma}(t)\sigma(t)\nabla_{\boldsymbol{z}_t} \log p(\boldsymbol{z}_t)\mathrm{d}t + \sqrt{\dot{\sigma}(t)\sigma(t)}\mathrm{d}\boldsymbol{\omega}_t.
\end{aligned}
\tag{39}
$$

The reverse imputation from time $t_i$ to $t_{i-1}$ also involves resampling in order to reduce the error brought by each step (Lugmayr et al., 2022). Resampling indicates that Eq. 39 will be repeated for $U$ steps from $\boldsymbol{z}_{t_i}$ to $\boldsymbol{z}_{t_{i-1}}$. After completing the reverse steps, we obtain the imputed latent vector $\boldsymbol{z}_0$, which could be put into TABSYN's VAE decoder to recover the original input data.

The algorithm for missing value imputation is presented in Algorithm 4.

---

**Algorithm 4:** TABSYN for Missing Value Imputation

---

1: **1. VAE encoding**
2: $\boldsymbol{z} = [\boldsymbol{z}^{\text{num}}, \boldsymbol{z}_1^{\text{oh}}, \cdots \boldsymbol{z}_{M_{\text{cat}}}^{\text{oh}}]$ is a data sample having missing values.
3: $m_{\text{num}}$ denotes the missing numerical columns.
4: $m_{\text{cat}}$ denotes the missing categorical columns.
5: $x_{:,j}^{\text{num}} \Leftarrow \text{mean}(\boldsymbol{z}_{:,j}^{\text{num}}), \forall j \in m_{\text{num}}$.
6: $\boldsymbol{z}_{i,j}^{\text{oh}} \Leftarrow [\frac{1}{C_j}, \cdots, \frac{1}{C_j}, \cdots \frac{1}{C_j}] \in \mathbb{R}^{1 \times C_j}, \forall j \in m_{\text{cat}}$.
7: $\boldsymbol{z} = \text{Flatten}(\text{Encoder}(\boldsymbol{z})) \in \mathbb{R}^{1 \times Md}$
8:
9: **2. Obtaining the masking vector**
10: $\boldsymbol{m} \in \mathbb{R}^{1 \times Md}$
11: **for** $j = 1, \ldots, Md$ **do**
12:    **if** $(\lfloor j/d \rfloor) \in m_{\text{num}}$ or $(\lfloor j/d \rfloor - M_{\text{num}}) \in m_{\text{cat}}$ **then**
13:       $\boldsymbol{m}_j = 0$
14:    **else**
15:       $\boldsymbol{m}_j = 1$
16:    **end if**
17: **end for**
18:
19: **3. Missing Value Imputation via Denoising**
20: $\boldsymbol{z}_{t_{\max}} = \boldsymbol{z} + \sigma(t_{\max})\boldsymbol{\varepsilon}, \boldsymbol{\varepsilon} \sim \mathcal{N}(\boldsymbol{0}, \boldsymbol{I})$
21: **for** $i = \max, \cdots, 1$ **do**
22:    **for** $u = 1, \cdots, U$ **do**
23:       $\boldsymbol{z}_{t_{i-1}}^{\text{known}} = \boldsymbol{z} + \sigma(t_{i-1})\boldsymbol{\varepsilon}, \boldsymbol{\varepsilon} \sim \mathcal{N}(\boldsymbol{0}, \boldsymbol{I})$
24:       $\boldsymbol{z}_{t_{i-1}}^{\text{unknown}} = \boldsymbol{z}_{t_i} + \int_{t_i}^{t_{i-1}} \mathrm{d}\boldsymbol{z}_{t_i}$
25:       $\mathrm{d}\boldsymbol{z}$ is defined in Eq. 6
26:       $\boldsymbol{z}_{t_{i-1}} = \boldsymbol{m} \odot \boldsymbol{z}_{t_{i-1}}^{\text{known}} + (1 - \boldsymbol{m}) \odot \boldsymbol{z}_{t_{i-1}}^{\text{unknown}}$
27:       **if** $u < U$ and $t > 1$ **then**
28:          ### Resampling
29:          $\boldsymbol{z}_{t_i} = \boldsymbol{z}_{t_{i-1}} + \sqrt{\sigma^2(t_i) - \sigma^2(t_{i-1})}\boldsymbol{\varepsilon}, \boldsymbol{\varepsilon} \sim \mathcal{N}(\boldsymbol{0}, \boldsymbol{I})$
30:       **end if**
31:    **end for**
32: **end for**
33: $\boldsymbol{z}_0 = \boldsymbol{z}_{t_0}$
34:
35: **4. VAE decoding**
36: $\hat{\boldsymbol{z}} = \text{Decoder}(\text{Unflatten}(\boldsymbol{z}_0))$
37: $\hat{\boldsymbol{z}}$ is the imputed data

---

**Classification/Regression as missing value imputation.** With Algorithm 4, we are able to use TABSYN for imputation with any missing columns. In this section, we consider a more interesting application – treating classification/regression as missing value imputation tasks directly. As illustrated in Section E.1, each dataset is assigned a machine learning task, either a classification or regression on the target column in the dataset. Therefore, we can mask the values of the target columns, then apply TABSYN to impute the masked values, completing the classification or regression tasks.

In Table 12, we present TABSYN's performance in missing value imputation tasks on the target column of each dataset. The performance is compared with directly training a classifier/regressor, using the remaining columns to predict the target column (the 'Real' row in Machine Learning Efficiency tasks, Table 3). Surprisingly, imputing with TabSyn shows highly competitive results on all datasets. On four of six datasets, TabSyn outperforms training a discriminate ML classifier/regressor on real data. We suspect that the reason for this phenomenon may be that discriminative ML models are more prone to overfitting on the training set. In contrast, by learning the smooth distribution of the entire data, generative models significantly reduce the risk of overfitting. The excellent results on the missing value imputation task further highlight the significant importance of our proposed TabSyn for real-world applications.

Our TABSYN is not trained conditionally on other columns for the missing value imputation tasks, and the performance can be further improved by training a separate conditional model specifically for this task. We leave it for future work.

Table 12: Performance of TABSYN in Missing Value Imputation tasks, compared with training an XGBoost classifier/regressor using the real data.

| Methods | Adult | Default | Shoppers | Magic | Beijing | News |
|---|---|---|---|---|---|---|
| | AUC ↑ | AUC ↑ | AUC ↑ | AUC ↑ | RMSE ↓ | RMSE ↓ |
| Real with XGBoost | 92.7 | 77.0 | 92.6 | **94.6** | 0.423 | **0.842** |
| Impute with TABSYN | **93.2** | **87.2** | **96.6** | 88.8 | **0.258** | 1.253 |

### F.5   IMPACTS OF THE QUALITY OF VAES

Intuitively, the performance of TabSyn appears to be highly dependent on the quality of the pre-trained VAE model. Therefore, we conduct further research to investigate how a less trained VAE model would impact the quality of synthetic data generated by TabSyn. To this end, we investigate the quality of synthetic data generated by TabSyn using the embeddings of the VAE obtained at different epochs as the latent space. Figure 9 plots the results of single-column density estimation and pair-wise column correlation estimation on the Adult and Default datasets, with intervals set at 400 epochs. We can observe that increasing the training epochs of the VAE does indeed improve the quality of TabSyn's generated data. Additionally, even when the VAE is sub-optimal (e.g., training epochs around 2000), TabSyn's performance is already very close to the optimal ones. Furthermore, even with a relatively low number of VAE training epochs (e.g., 800-1200), TabSyn's performance approaches or even surpasses the most competitive baseline, TabDDPM. Based on this observation, we recommend thoroughly training the VAE to achieve superior data generation quality when resources are abundant. However, when resources are limited, reducing the VAE training duration still yields decent performance.

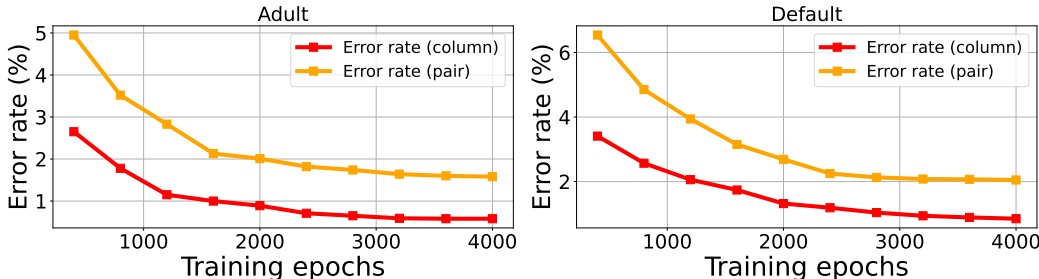

Figure 9: Performance of TABSYN with VAEs trained for different epochs. By default, TABSYN's VAE is trained with 4000 epochs

### F.6   PRIVACY PROTECTION: DISTANCE TO CLOSEST RECORD (DCR)

To study if the synthetic data will cause privacy information leakage issues, we compute the DCRs of the synthetic data. Specifically, we follow the 'synthetic vs. holdout' setting [10]. We initially divide the dataset into two equal parts: the first part served as the training set for training our generative model, while the second part was designated as the holdout set, which is not used for training. After completing model training, we sample a synthetic set of the same size as the training set (and the holdout set).

---

[10] https://www.clearbox.ai/blog/2022-06-07-synthetic-data-for-privacy-\preservation-part-2

We then calculate the DCR scores for each sample in the synthetic set concerning both the training set and the holdout set. We can create histograms to visualize the distribution of DCR scores for the synthetic set in comparison to both the training and holdout sets. Intuitively, if there is a privacy issue (e.g. if the synthetic set is directly copied from the training set), then the DCR scores for the training set should be closer to 0 than those for the testing set. Conversely, if there is no privacy issue, the distributions of DCR scores of the training and holdout sets should largely overlap. In Figure 10, we plot the distributions of synthetic sets' DCRs concerning the training set and holdout set on Default and Shoppers. We can observe that deep generative models such as CoDi, STaSy, TabDDPM, and TabSyn do not suffer from privacy issues, while the interpolation-based method SMOTE might not be able to protect privacy information.

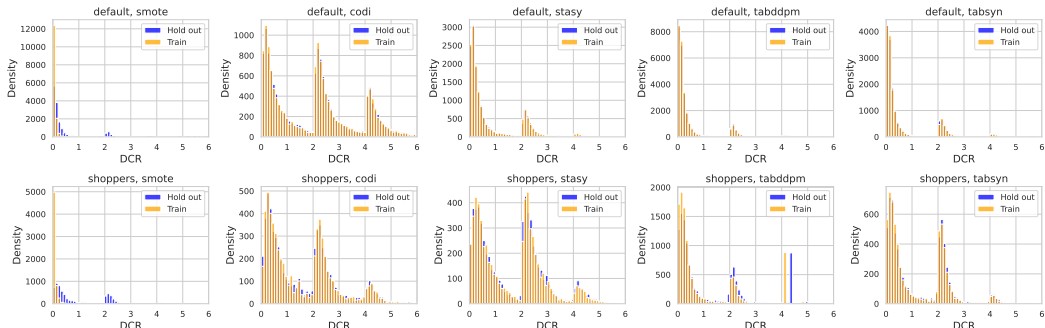

Figure 10: Distributions of the DCR scores between the synthetic dataset and the training/holdout datasets. Deep generative models have similar DCR distributions concerning the training set and holdout set, while in the interpolation-based method SMOTE, DCRs concerning the training set are much smaller than DCRs concerning the holdout set.

Additionally, we calculate the probability that a synthetic sample is closer to the training set (rather than the holdout set). When this probability is close to 50% (i.e., 0.5), it indicates that the distribution of distances between synthetic and training instances is very similar (or at least not systematically smaller) than the distribution of distances between synthetic and holdout instances, which is a positive indicator in terms of privacy risk. Table 13 displays the results obtained by different models on Default and Shoppers datasets.

Table 13: DCR score (the probability that a synthetic example is closer to the training set rather than the holdout set ($\%$, a score closer to $50\%$ is better).

| Method | Default | Shoppers |
|---|---|---|
| SMOTE | $91.41\%_{\pm 3.42}$ | $96.40\%_{\pm 4.70}$ |
| STaSy | $50.23\%_{\pm 0.09}$ | $51.53\%_{\pm 0.16}$ |
| CoDi | $51.82\%_{\pm 0.26}$ | $51.06\%_{\pm 0.18}$ |
| TabDDPM | $52.15\%_{\pm 0.20}$ | $63.23\%_{\pm 0.25}$ |
| TABSYN | $51.20\%_{\pm 0.18}$ | $52.90\%_{\pm 0.22}$ |

## G  DETAILS FOR REPRODUCTION

In this section, we introduce the details of TABSYN, such as the data preprocessing, training, and hyperparameter details. We also present details of our reproduction for the baseline methods.

### G.1  DETAILS OF IMPLEMENTING TABSYN

**Data Preprocessing.**  Real-world tabular data often contain missing values, and each column's data may have distinct scales. Therefore, we need to preprocess the data. Following TabDDPM (Kotelnikov et al., 2023), missing cells are filled with the column's average for numerical columns. For

categorical columns, missing cells are treated as one additional category. Then, each numerical/categorical column is transformed using the QuantileTransformer[11]/OneHotEncoder[12] from scikit-learn, respectively.

**Hyperparameter settings.** TABSYN **uses the same set of parameters for different datasets.** (except $\beta_{\max}$ for Shoppers). The detailed architectures of the VAE and Diffusion model of TABSYN have been presented in Appendix D.1 and Appendix D.2, respectively. Below is the detailed hyperparameter setting.

Hyperparameters of VAE:

- Token dimension $d = 4$,
- Number of Layers of VAEs' encoder/decoder: 2,
- Hidden dimension of Transformer's FFN: $4 * 32 = 128$,
- $\beta_{\max} = 0.01$ ($\beta_{\max} = 0.001$ for Shoppers),
- $\beta_{\min} = 10^{-5}$,
- $\lambda = 0.7$.

Hyperparameters of Diffusion:

- MLP's hidden dimension $d_{\text{hidden}} = 1024$.

Unlike the cumbersome hyperparameter search process in some current methods (Kotelnikov et al., 2023; Kim et al., 2023; Lee et al., 2023) to obtain the optimal hyperparameters, TABSYN consistently generates high-quality data without the need for meticulous hyperparameter tuning.

### G.2 DETAILS OF IMPLEMENTING BASELINES

Since different methods have adopted distinct neural network architectures, it is inappropriate to compare the performance of different methods using identical networks. For fair comparison, we adjust the hidden dimensions of different methods, ensuring that the numbers of trainable parameters are close (around 10 million). Note that enlarging the model size does lead to better performance for the baseline methods. Under these conditions, we reproduced the baseline methods based on the official codes, and **our reproduced codes are provided in the supplementary**. Below are the detailed implementations of the baselines.

**CTGAN and TVAE** (Xu et al., 2019): For CTGAN, we follow the implementations in the official codes[13], where the hyperparameters are well-given. Since the default discriminator/generator MLPs are small, we enlarge them to be the same size as TABSYN for fair comparison. The interface for TVAE is not provided, so we simply use the default hyperparameters defined in the TVAE module. The sizes of TVAE's encoder/decoder are enlarged as well.

**GOGGLE** (Liu et al., 2023b): We follow the official implementations[14]. In GOGGLE's official implementation, each node is a column, and the node feature is the 1-dimensional numerical value of this column. However, GOGGLE did not illustrate and failed to explain how to handle categorical columns [15]. To extend GOGGLE to mixed-type tabular data, we first transform each categorical column into its $C$-dimensional one-hot encoding. Then, each single dimension of $0/1$ binary values becomes the graph node. Consequently, for a mixed-type tabular data of $M_{\text{num}}$ numerical columns and $M_{\text{cat}}$ categorical columns and the $i$-th categorical column of $C_i$ categories, GOGGLE's graph has $M_{\text{num}} + \sum_i C_i$ nodes.

---

[11]https://scikit-learn.org/stable/modules/generated/sklearn.preprocessing.QuantileTransformer.html
[12]https://scikit-learn.org/stable/modules/generated/sklearn.preprocessing.OneHotEncoder.html
[13]https://github.com/sdv-dev/CTGAN
[14]https://github.com/vanderschaarlab/GOGGLE
[15]https://github.com/tennisonliu/GOGGLE/issues/2

**GReaT**: We follow the official implementations[16]. During our reproduction, we found that the training of GReaT is memory and time-consuming (because it is fine-tuning a large language model). Typically, the batch size is limited to 32 on the Adult dataset, and training for 200 epochs takes over 2 hours. In addition, since GReaT is textual-based, the generated content is not guaranteed to follow the format of the given table. Therefore, additional post-processing has to be applied.

**STaSy** (Kim et al., 2023): In STaSy, the categorical columns are encoded with one-hot encoding and then are put into the continuous diffusion model together with the numerical columns. We follow the default hyperparameters given by the official codes[17] except for the denoising function's size, which is enlarged for a fair comparison.

**CoDi** (Lee et al., 2023): We follow the default hyperparameters given by the official codes[18]. Similarly, the denoising U-Nets used by CoDi are also enlarged to ensure similar model parameters.

**TabDDPM** (Kotelnikov et al., 2023): The official code of TabDDPM[19] is for conditional generation tasks, where the non-target columns are generated conditioned on the target column(s). We slightly modify the code to be applied to unconditional generation.

---

[16]https://github.com/kathrinse/be_great/tree/main
[17]https://github.com/JayoungKim408/STaSy/tree/main
[18]https://github.com/ChaejeongLee/CoDi/tree/main
[19]https://github.com/yandex-research/tab-ddpm

