# OpenReview forum: "Mixed-Type Tabular Data Synthesis with Score-based Diffusion in Latent Space"
_ICLR.cc/2024/Conference — ICLR 2024 oral_

### Official Review · Reviewer_dVGm · 2023-10-30

**Soundness:** 3 good
**Presentation:** 4 excellent
**Contribution:** 3 good
**Rating:** 8
**Confidence:** 4

**Summary:**

This paper present a new latent diffusion model/code for tabular data generation.
Transposing to tabular data the recent ideas of (Rombach et al. CVPR 2022 and Karras et al. NeuIPS 2022), their model architecture is two-folds:
- a transformer-based \beta-VAE to embedd tabular data into a latent space
- a score-based generator based on (Song et al. ICLR 2021)'s architecture

As a slight algorithmic contributions, the authors propose to use an "adaptive VAE loss weighing".

In the experiment section, the method is benchmarked against 6 state of the art tabular data generation models on 6 datasets. A few ablation tests are provided (one to justify the adaptive weighing).
This paper is only focused on unconditional generation, some experiments on missing-values imputation are also provided.

It is worth noting that both the code and a rich appendix are provided as supplementary material.
The code is clear and well commented.
The appendix provides a clear background on recent score-based generation best-practices and several supplementary experiments. Several implementation and methodology details allows the readers to retrieve what they needs to reproduce and understand this work.

**Strengths:**

I suggest acceptance:

- I really liked reading this paper. It is well written with several clear illustrations.
- The contribution is mostly incremental but solid and well driven.
- The provided code is clear and will be useful for the community (if it is published)
- The supplementary material provides a detailed and clear background summary.
- The experiment section could be improved but seems solid: the method seems efficient and fast when compared against other SOTA methods.
- The model is tested with a single hyper-parameter configuration on all datasets

**Weaknesses:**

- The scientific contribution is mostly incremental and expected
- The authors claim that no "unified and comprehensive evaluation" exists for tabular data synthesis. To my opinion, one weakness of this paper is indeed that it feeds this lack of a unified benchmark by proposing another new benchmark with new metrics that are not used in other papers. Sticking a bit more to previous paper's metrics and datasets could improve that point.
- given the size of the appendix, one is surprised to see that no simple baselines like Bayesian Networks or SMOTE are provided in the experiments. SMOTE is known to be a competitive baseline for "target-conditional" data generation.
- No privacy preservation metrics (like DCR) are provided. No detection test metric (like C2ST) is provided
- The absence of hyper-parameters tuning in the benchmark is both laudable and questionable as it may hinder some of the other models (a fair option could be to report the total training time with a fixed budget).

**Questions:**

- Could you use the "sdmetrics" library to provide some privacy (like DCR) and detection (like C2ST) metrics in your benchmark ?
- A few more datasets common with previous papers like (Kotelnikov et al. 2022, see Table 2) could improve the benchmark.
- Could you add SMOTE (with unconditional sampling) as a baseline in your results ?
- Are the confidence intervals on result tables computed through cross validation or only through multiple-sampling ?

- Will you publish your code ?
- Did you try your code on 2d synthetic sklearn examples ?

---

> ### Author Response · Authors · 2023-11-17
> **Response to Reviewer dVGm (1/2)**
>
> We appreciate the insightful feedback from the reviewer, which improves our evaluation setup.
>
> > Q1 & W4 Provide privacy metrics (e.g., DCR) and detection metrics (e.g., C2ST) in the benchmark.
>
> Based on the suggestion, we have included the DCR score as a metric for assessing privacy protection, as well as C2ST as a metric for detection. Below are the implementation details of these two metrics and our preliminary results.
>
> **For the DCR score**, it appears that the “sdmetrics” library does not provide a corresponding implementation. Therefore, we developed this metric from scratch. Specifically, we followed the 'synthetic vs. holdout' setting as described in 'https://www.clearbox.ai/blog/2022-06-07-synthetic-data-for-privacy-preservation-part-2.' We initially divided the dataset into two equal parts: the first part served as the training set for training our generative model, while the second part was designated as the holdout set, which was not used for training. After the completion of model training, we sampled a synthetic set of the same size as the training set (and the holdout set).
>
> We then calculated the DCR scores for each sample in the synthetic set with respect to both the training set and the holdout set. We can create histograms to visualize the distribution of DCR scores for the synthetic set in comparison to both the training and holdout sets. Intuitively, if there is a privacy issue (e.g., if the synthetic set is directly copied from the training set), then the DCR scores for the training set should be closer to 0 than those for the testing set. Conversely, if there is no privacy issue, the distribution of DCR scores for the training and holdout sets should largely overlap.  We plot these figures and have updated them in Figure 10, Appendix F.5 in the revised paper.
>
> Additionally, we can calculate the probability that a synthetic sample is closer to the training set (rather than the holdout set). If this probability is close to 50% (i.e., 0.5), it indicates that the distribution of distances between synthetic and training instances is very similar (or at least not systematically smaller) than the distribution of distances between synthetic and holdout instances. This finding is a positive indicator in terms of privacy risk. The table below displays the results obtained by different models, **including SMOTE**, on this metric, on Default and Shoppers datasets:
>
>
> | Method        | Default            | Shoppers    |
> | --------      | -------            | -------     |
> | SMOTE         |  91.41%±3.42       | 96.40%±4.70 |
> | STaSy         |  50.23%±0.09       | 51.53%±0.16 |
> | Codi          |  51.82%±0.26       | 51.06%±0.18 |
> | TabDDPM       |  52.15%±0.20       | 63.23%±0.25 |
> | TabSyn (ours) |  51.20%±0.18       | 52.90%±0.22 |
>
>
> **For the C2ST score**, we employed the detection metric provided by sdmetrics: https://docs.sdv.dev/sdmetrics/metrics/metrics-in-beta/detection-single-table. This metric assesses how challenging it is to distinguish real data from synthetic data. We utilized the built-in logistic regression and SVC detectors for evaluation. In the table below, we present the results obtained using logistic regression as the detection method.
>
> | Method | Adult | Default | Shoppers | Magic  | Beijing | News |
> | --- | --- | --- | --- | --- | --- | --- |
> |SMOTE | 0.9710 | 0.9274 | 0.9086 | 0.9961 | 0.9888 |  0.9344|
> | CTGAN | 0.5949 | 0.4875 | 0.7488 | 0.6728 | 0.7531 | 0.6947 |
> | TVAE | 0.6315 | 0.6547 | 0.2962 | 0.7706 | 0.8659 | 0.4076 |
> | GOGGLE | 0.1114 | 0.5163 | 0.1418 | 0.9526 | 0.4779 | 0.0745 |
> | GReaT | 0.5376 | 0.4710 | 0.4285 | 0.4326 | 0.6893 | - |
> | STaSy | 0.4054 | 0.6814 | 0.5482 | 0.6939 | 0.7922 | 0.5287 |
> | CoDi | 0.2077 | 0.4595 | 0.2784 | 0.7206 | 0.7177 | 0.0201 |
> | TabDDPM | 0.9755 | 0.9712 | 0.8349 | **0.9998**| 0.9513 | 0.0002 |
> | TabSyn | **0.9986** | **0.9870** | **0.9740** | 0.9732 |**0.9603** | **0.9749** |
>
> As indicated in the table, the Detection score exhibits superior discriminative power compared to other metrics such as single-column density estimation, pair-wise column shape estimation, and MLE. The detection score shows significant variations across different models for synthetic data generation. The proposed TabSyn consistently achieves notably high scores across all datasets. SMOTE directly interpolates within the training set, so it is not surprising that it achieves high scores in the detection metric.
>
> We've updated these results in the revised paper in Appendix F.5 and AppendiX F.6. In the latest version of the code, we have added these two metrics to evaluate the performance of synthetic data on privacy protection and detection tasks. Please refer to the updated README.md file for details. Thanks again for the suggestions.

---

> > ### Comment · Reviewer_dVGm · 2023-11-17
> > **Q1 & W4 Provide privacy metrics (e.g., DCR) and detection metrics (e.g., C2ST) in the benchmark.**
> >
> > You are right, dcr and dcr_rate are not provided in sdmetrics. It should be as it is simple to implement.
> > I appreciate, this quick improvement. It shows that TabSyn is both safe and realistic.

---

> ### Author Response · Authors · 2023-11-17
> **Response to Reviewer dVGm (2/2)**
>
> > Q2 & W2: Incorporate more datasets.
>
> We selected datasets for our research based on two key criteria: data type diversity and accessibility. Since our study focuses on synthesizing tabular data containing both numerical and categorical features, we needed datasets with mixed data types. Also we preferred openly available datasets that could be easily downloaded for reproducibility.  After reviewing options, we chose six representative tabular datasets with mixed data types from UCI Machine Learning Repository to use in our experiments. The datasets contain a combination of numerical and categorical features relevant to our research goals. To facilitate future work, we provide code in our open-source codebase to download these datasets.
>
> We recognize that many datasets in Table 2 of TabDDPM (Kotelnikov et al. 2022) only include numerical features, which do not match the mixed data types needed for this research. To expand the availability of suitable datasets, we plan to acquire and integrate additional datasets from Table 2 that do contain both numerical and categorical features. We welcome contributions from the research community to further expand the selection of datasets compatible with our codebase and support new lines of research.
>
>
> > Q3 & W3: Add SMOTE as a baseline in the results.
>
> Thank you for the reviewer's suggestion; indeed, SMOTE can also be utilized for synthesizing new data in tabular data scenarios. Our initial idea was to investigate the synthesis of new data using deep generative models from random noise. While SMOTE can also generate new data, being an interpolation-based method, it may lack a certain level of randomness. Furthermore, SMOTE may encounter certain challenges when it comes to interpolating categorical features. Therefore, we did not consider it as a baseline in our study.
>
> Since the reviewer suggested it, we conducted experiments using SMOTE on the six datasets studied in this paper. We transformed categorical features into one-hot encoding for interpolation purposes. The table below shows the performance of SMOTE on various tasks across different datasets (Furthermore, in the previous response, we also provided its performance in terms of privacy protection and detection):
>
> | Metric | Adult | Default | Shoppers | Magic  | Beijing | News |
> | --- | --- | --- | --- | --- | --- | --- |
> | Single Column | 1.6\% | 1.48\% | 2.68\% | 0.91\% | 1.85\% | 5.31\% |
> | Pair Correlation | 3.28\% | 8.41\% | 3.56\% | 3.16\% | 2.39\% | 5.38\% |
> | MLE | 0.899 | 0.741 | 0.911 | 0.934 | 0.593 | 0.897 |
>
> We note that by interpolating on individual dimensions, SMOTE exhibits excellent performance in the single-column density estimation metric. However, in the column pair correlation estimation metric, SMOTE's performance becomes suboptimal. This could be attributed to the possibility of non-linear correlations between different columns, making it challenging to reconstruct them through interpolation. On the Machine Learning Efficiency task as well, the data generated by SMOTE has achieved commendable performance. However, as discussed in Q1 & W4, the DCR score of the data generated by SMOTE is notably low, indicating a significant probability of direct copying from the training set.
>
> Therefore, we believe that SMOTE can serve as a data augmentation method but may not be suitable as a standalone generative model. In the latest version of the code, we have added support for SMOTE as a new baseline. Please check the updated code if you are interested. We will also include the results above in the corresponding tables in the revised table.
>
> > Q4: Confidence Intervals.
>
> The confidence score is computed through cross-validation. To be detailed, we randomly split the dataset 20 times, and within each split we train the model and sample the synthetic dataset.
>
>
> > Q5: Publishment of the codes.
>
> Sure, we will publish the entire codebase.
>
>
> > Q6: Try the codes on 2d synthetic sklearn examples.
>
> Our method can also be applied to 2d synthetic data from sklearn. In the updated codes, we have added a 2d synthetic dataset 'blob' of three classes created using sklearn.makeblobs. It is reformulated as a tabular dataset of 3 columns, where the first two columns are the 2d numerical features, and the last column is the categorical feature, denoting the label of each example. We also created a notebook blob.ipynb for facilitating running experiments on the synthetic dataset.

---

> > ### Comment · Reviewer_dVGm · 2023-11-17
> > **Q3 & W3: Add SMOTE as a baseline in the results.**
> >
> > Great. SMOTE is indeed good for MLE but poor at preserving privacy, but it is simple and cheap to train: it hence makes a sound baseline for tabular generative models.

---

### Official Review · Reviewer_aVor · 2023-10-30

**Soundness:** 4 excellent
**Presentation:** 4 excellent
**Contribution:** 2 fair
**Rating:** 6
**Confidence:** 4

**Summary:**

Authors propose a generative model for mixed type tabular data. The proposed model first tokenizes the mixed type columns, feeds it to a one transformer layer, which then forms as the encoder in the VAE model. Finally the latent space is fixed by diffusion.

**Strengths:**

Very reasonable model for the mixed type tabular data. Definitely something that I would use in my day to day work. Results are also convincing.

**Weaknesses:**

- Model itself seems to be pretty much the same as Vahdat 2021, except that in that paper authors used only images, whereas now tokenization is needed to use the same model. I would like authors to comment on this, and it would really help the paper to be very clear in the Introduction that where the technical novelty lies.

**Questions:**

- what would be the accuracy in the downstream task if latent code would be used directly (and no synthetic data). I understand that this is not possible for all models. But for the models that it is possible it would be interesting to see how much benefit there is (i.e. can you win real)
- Are all classification tasks in downstream binary tasks? If no, then AUC is not a correct metric.

---

> ### Author Response · Authors · 2023-11-17
> **Response to Reviewer aVor (1/2)**
>
> Thank you for your insightful comments and suggestions to help us improve the clarity and soundness of our research. Every question raised by you has been thoughtfully examined.
>
> > W1: The difference between Latent Score-based Generative
> Model (LSGM) and our proposed TabSyn, and the technical novelty
>
>
> We acknowledge your observation regarding the similarity between Tabsyn and LSGM (Vahdat 2021). However, it is important to highlight key distinctions.
>
> 1. A key difference between tabular data and image data lies in the variety of data types present in tabular data, encompassing both numerical and categorical features. In contrast, image data primarily comprises continuous pixel values that denote color and intensity. Tabsyn is specifically designed for mixed-type tabular data, a domain with unique challenges not addressed by LSGM. The feature tokenization process in Tabsyn represents a notable advancement tailored for such data, ensuring effective handling of diverse data types (numeric, categorical, etc.).
> 2. Additionally, a fundamental aspect of tabular data is its complex relationships between columns, which differs from the localized spatial correlation observed in images, where pixel values largely correlate with adjacent pixels. Tabsyn uses a lightweight transformer model as the encoder and decoder in the VAE to capture intricate columns relationships. The self-attention mechanism in the Transformer enables each column to dynamically interact and integrate information from all other columns. This is accomplished by learning the importance weights of every other columns when processing a particular column.
> 3. Training a VAE on tabular data presents a distinct challenge compared to training on image data. With image data, small errors in pixel values often do not significantly impact object recognition. However, with tabular data, precision in the values of each column is relatively more important for preserving the semantics of the data. As a result, a higher weight on the reconstruction loss is beneficial when training a VAE on tabular data in order to minimize distortion of the input. However, an excessively high reconstruction loss weight can lead to a poor approximation of the posterior distribution, $q(z)$. To address this issue, we introduce an adaptive schedule for the weight parameter, $\beta$, which allows more effective training of the VAE model on tabular data. The rationale and efficacy of this adaptive weight schedule are demonstrated through the results presented in Figure 3 and Table 4 in the paper.
> 4. Although the diffusion process is conceptually similar to LSGM, we make further improvements to the sampling speed. Specifically, we introduce a noise level proportional to the time step. Through theoretical analysis and empirical experiments, we demonstrate this improvement significantly reduces the number of steps needed to generate synthetic data compared to LSGM.
>
> Overall, Tabsyn's technical innovations lie in its specialized approach to handling the mixed-type and complexity of tabular data, from (1) its feature tokenization process and (2) its sophisticated use of a transformer model to capture column relationships to (3) its adaptive training schedule for VAE and (4) enhanced diffusion process for efficient data generation. These technical advancements are critical in enhancing the model's performance and are not present in LSGM.

---

> ### Author Response · Authors · 2023-11-17
> **Response to Reviewer aVor (2/2)**
>
> > Q1: The accuracy in the downstream tasks when the latent codes are directly used.
>
> Our paper focuses on synthesizing mixed-type tabular data using deep generative models. The primary goal is to generate synthetic data that faithfully reproduces the distribution of the original dataset. The variational autoencoder (VAE) in TabSyn learns latent representations of each row of tabular data, but is not optimized for any specific downstream task.
>
> Most importantly, since we currently focus on unconditional training, label information (such as class labels in classification tasks and target values in regression tasks) is trained and generated alongside other columns' features. Therefore, the latent encoding obtained by the VAE model already contains label information. In this case, if we directly use latent encoding to predict downstream tasks, there is actually a risk of label leakage. We conducted experiments in this regard and found that accuracy, AUC score, and other metrics were all close to 100%.
>
>
> > Q2: Are all classification tasks in downstream binary tasks? (AUC)
>
> Yes, the four classification datasets - Adult, Default, Shoppers and Magic - are all for binary classification. As a result, we use the AUC score as the metric in Machine Learning Efficiency tasks.
>
> Our proposed method can also directly train on multi-class classification datasets without requiring modifications during training since it is an unconditional generation approach. It simply requires additional metrics to evaluate the downstream multi-class classification task. As suggested by Reviewer dVGm, we created a synthetic multi-class classification dataset. We also provided a blob.ipynb notebook to facilitate creating such a dataset. Please check the updated codes if you are interested (https://anonymous.4open.science/r/TabSyn-ICLR-Submission-6938/blob.ipynb).

---

> ### Author Response · Authors · 2023-11-21
> **A reminder**
>
> Dear reviewer, we have submitted my reply a few days ago. Now the deadline for reviewer-author discussion is approaching, but we observe that you have not replied to my comment. We would be happy if you let me know if you still have some questions or replies. If so, we will reply promptly. Looking forward to your reply.

---

> > ### Comment · Reviewer_aVor · 2023-11-22
> > **Reviewer answer**
> >
> > I apreciate authors answer and clarification of the technical novelty. I am willing to raise the score.

---

### Official Review · Reviewer_Fs1y · 2023-10-31

**Soundness:** 3 good
**Presentation:** 3 good
**Contribution:** 4 excellent
**Rating:** 8
**Confidence:** 3

**Summary:**

Extending diffusion models to handle tabular data presents challenges due to the complex distributions and diverse data types inherent to such data. To address this, the authors introduce the use of a Variational Autoencoder (VAE) to learn a regularized latent embedding representation of the data, which is subsequently processed by a diffusion network for synthesis. Notably, the study employs a comprehensive set of multi-dimensional evaluation metrics for the generated data, filling a gap often observed in previous research. The proposed method excels across these metrics, underscoring its efficacy in generating synthetic data that closely mirrors the original data distribution.

**Strengths:**

- The method effectively manages mixed-type data by transforming them into a single cohesive space to ensure capturing or inter-column relationships.

- Compared to existing diffusion-based methods, this method requires fewer reverse steps and offers faster data synthesis.

- The authors have provided a unified comparison environment for their proposed tabular data synthesis, as well as all the compared baseline methods, and made their code base publicly available.

- The study employs a diverse set of multi-dimensional evaluation metrics for a holistic assessment of the generated data, addressing a common shortcoming in previous research.

- The method has been rigorously tested on six datasets using five metrics, and it consistently outperforming other existing methods, indicating its prowess in generating synthetic data that closely reflects the original data distribution.

**Weaknesses:**

- The method's efficacy is contingent upon a well-trained VAE. It would be beneficial to compare the outcomes between optimally and sub-optimally trained VAEs, providing insights into worst-case vs. best-case scenarios.

- Given the generative capability of VAEs, it would be insightful to see results from data generated solely by the VAE used in this study. The distinction between the paper's transformer-based VAE and TVAE warrants further exploration to determine the independent efficacy of the former.

- While adjusting default hyperparameters for a fair comparison is commendable, understanding performance under default settings across consistent training epochs would give a fuller picture. This would ascertain whether hyperparameter enlargement (as done for CTGAN and CoDi) equally benefits the models or favors the presented method disproportionately.

- The discrepancy observed where TabDDPM struggles with the News dataset (poor performance in Table 1), yet exhibits a low error rate in Table 2 seems unintuitive. Additionally, given TabDDPM's consistent second-place ranking, except for the News dataset, its fourth-place average rank seems unfair. An alternative could be per-dataset ranking or reporting modal / averaged ranks.

- Given that the News dataset is primarily of numeric nature, it seems counterintuitive that TabDDPM, a diffusion based model would underperform on this dataset. It would be beneficial to understand the authors' rationale behind the model's inability to generate meaningful content for this dataset.

- The deployment of MLE as a metric for privacy is unconventional. Traditionally, MLE assesses the synthetic data's task-performance equivalence to real data, not privacy leakage. It would be enriching if the authors could shed light on this choice.


Overall, this paper stands out for its meticulous code, articulate presentation, and thorough analysis. I commend the authors for their contribution.

**Questions:**

Thank you for sharing your code with the community; it's a valuable resource. While exploring it, I encountered a few queries and points of feedback:

1. **Device Attribute Error**: When executing the command `python main.py —dataname adult —method vae —mode train`, I came across the “AttributeError: ’Namespace’ object has no attribute ‘device’”. I was able to address this by introducing an else statement post line 7 in `main.py` to default to 'cpu'. Consider incorporating this for broader compatibility.
   ```python
   if …:
       args.device = …
   else:
       args.device = 'cpu'
   ```

2. **Sample Size Limitation**: I attempted the VAE training phase with 40 samples and encountered an `IndexError: index out of range in self`. This wasn't an issue with the full sample size of 32561. Is the model designed to accommodate only larger samples, or is there a potential to adapt it to smaller sample sizes?

3. **Epoch Setting for VAE Model**: The default epoch for the VAE model in the code is set to 4000. Based on my prior experiences with the TVAE model using CTGAN's code, training for around 300 epochs usually suffices. Is the transformer architecture inherently more demanding in terms of training duration? Additionally, what criteria do you rely on to determine the termination of training? Introducing an early stopping mechanism might be beneficial, especially considering the subsequent training phase for the diffusion model. It's also noteworthy that a Train/Val accuracy of 100% seems achievable by the 1000th epoch.

---

> ### Author Response · Authors · 2023-11-17
> **Response to Reviewer Fs1y (1/3)**
>
> We appreciate the thoughtful feedback and suggestions from the reviewer on our paper. In the responses below, we have carefully addressed each of the reviewer's questions.
>
> > Q1: Device Attribute Error.
>
> Thanks for identifying this issue. We have addressed it in the latest codebase version.
>
> > Q2: Sample Size Limitation.
>
> We tried using a smaller batch size of 40, as suggested but did not see the error you described. Could you please provide more details on where exactly the error occurred, the exact command, and the experimental environment you used that triggered the issue? If you're using an older version of the code, please try downloading the latest version from the anonymous GitHub repository, which contains some new fixes.
>
> > Q3: Epoch setting for VAE Model.
>
> **Why TabSyn requires a larger number of training epochs?**
>
> On one hand, optimizing a Transformer can indeed be more challenging than an MLP, necessitating a greater number of epochs to ensure VAE convergence. On the other hand, our VAE requires the scheduling of the trade-off hyperparameter beta during training, which involves a certain number of steps. Depending on the training stage, we gradually adjust the importance weighting between Reconstruction loss and KL loss (see Figure 3 in the submitted paper).
>
> **Criteria used to determine the termination of training.**
>
> In all of the experiments, we trained the VAE for a fixed 4000 epochs. This decision was based on our observation that after training for 4000 epochs, the VAE's validation reconstruction loss had already approached convergence to a low value. This indicates that the learned VAE is capable of effectively reconstructing the input data. We did not employ any additional techniques to prematurely terminate the training of the VAE.
>
> **Introducing early stopping might be beneficial.**
> We have implemented early stopping in the updated codebase to terminate VAE training prematurely based on the suggestion to use this technique.
>
> **Train/Val accuracy of 100% seems achievable by the 1000th epoch.**
>
> The output training/validation accuracy solely reflects the reconstruction correctness of categorical features, and therefore, it cannot represent the overall reconstruction results since numerical features also require reconstruction. Moreover, consider a binary categorical variable where the ground truth is class 0. Predicted class distributions of [0.55, 0.45] and [0.99, 0.01] would both be considered correct predictions, but it's evident that we prefer predictions with higher confidence. Hence, Cross-Entropy loss is a more effective measure than accuracy in reflecting the quality of reconstruction, as it takes into account the confidence in predictions.
>
> > W1: Performance of sub-optimally trained VAEs.
>
> We investigated the quality of synthetic data generated by TabSyn using the embeddings of the VAE obtained at different epochs as the latent space. In the updated paper's Appendix F.4, Figure 9, we plot the results of single-column density estimation and pair-wise column correlation estimation on the Adult and Default datasets, with intervals set at 400 epochs. We can observe that
> 1. Increasing the training epochs of the VAE improves the quality of TabSyn's generated data.
> 2. even when the VAE is sub-optimal (e.g., training epochs around 2000), TabSyn's performance is close to the optimal ones.
> 3. even with a relatively low number of VAE training epochs (e.g., 800-1200), TabSyn's performance approaches or even surpasses the most competitive baseline, TabDDPM.
>
> Based on these observations, we recommend thoroughly training the VAE to achieve superior data generation quality when resources are abundant. However, when resources are limited, reducing the VAE training duration still yields decent performance.
>
> Besides, in Figure 6 of the initial paper submission, we investigated the sensitivity of hyperparameters in the VAE on TabSyn's performance. This can be considered as an evaluation of TabSyn's performance under the assumption of a suboptimal VAE.

---

> ### Author Response · Authors · 2023-11-17
> **Response to Reviewer Fs1y (2/3)**
>
> > W2: Synthetic data generated directly from TabSyn's VAE
>
> We studied the data synthesis capability of TabSyn's VAE under different $\beta$ settings just as done in Table 4. The results on Adult dataset concerning density estimation errors are presented in the following table.
>
> |  $\beta$           |  Single            | Pair         |
> | --------           | -------            | -------      |
> | VAE: $\beta=1.0$(Vanilla VAE) | 12.67\% $\pm$ 1.44 | 23.37\% $\pm$ 2.15 |
> | VAE: $\beta=0.1$              | 24.01\% $\pm$ 1.6 | 37.61\% $\pm$ 1.99 |
> | VAE: $\beta=0.01$             | 31.99\% $\pm$ 1.5 | 45.22\% $\pm$ 3.31 |
> | VAE: Scheduled $\beta$  | 36.25\% $\pm$ 2.26 | 50.18\% $\pm$ 3.19 |
> | TabSyn  | **0.58\%** $\pm$ 0.06 | **1.54\%** $\pm$ 0.27 |
>
> As demonstrated in the table, directly using TabSyn's VAE model to obtain synthetic data leads to poor data quality, regardless of the $\beta$.  Indeed, a small $\beta$ can lead to effective reconstruction of the original table, but at the cost of $q(z)$ deviating significantly from the standard normal distribution $\mathcal{N}(0, I)$. On the other hand, a larger beta can make $q(z)$ approach $N(0, I)$, but it may result in poorer reconstruction quality. Furthermore, we found that, for the VAE, maintaining $q(z)$ close to $\mathcal{N}(0, I)$ is advantageous for generating higher-quality synthetic data.
>
> > W3: Performance under different parameter scales
>
> We agree with the reviewer that comparing the performance of different methods of different scales of parameters will provide a broder comparison of each method. To this end, we adjust the hidden dimension of TabSyn's denoising MLP (the architecture of which is in Appendix D.2, Figure 8) within [128, 256, 512, 1024] (1024 is the default value). We adjust those for baseline methods CTGAN, CoDi and TabDDPM correspondingly. The following table presents the performance comparison concerning single column and pair correlation on Adult (singl-column error / pair correlation error):
>
> |   Hidden dim  | 128  | 256   | 512  | 1024   |
> | --------   | ------- | ------- |  ------- | ------- |
> |  CTGAN  | 23.14 / 28.92 | 20.64 / 25.26 |  17.92 / 22.89 | 16.84 / 20.23 |
> |  CoDi   | 29.16 / 29.65 | 26.42 / 27.58 | 22.91 / 24.06 | 21.38 / 22.49 |
> |  TabDDPM   | 8.69 / 12.39 | 5.36 / 8.75 | 2.32 / 4.19 | 1.75 / 3.01 |
> |  TabSyn | **4.02** / **8.51** | **1.92** / **4.22** | **0.65** / **2.06** | **0.58** / **1.54** |
>
> A clear observation is that when reducing the model's parameter count, all models exhibit a similar degree of performance decline. This is easily understood because when the model's capacity is not large enough, it becomes challenging to accurately learn the distribution of complex data. Our TabSyn maintains a significant advantage even when the parameter count is reduced.

---

> ### Author Response · Authors · 2023-11-17
> **Response to Reviewer Fs1y (3/3)**
>
> > W4 & W5: The strange behavior of TabDDPM on News dataset.
>
> The reviewer observes an interesting phenomenon with TabDDPM. When we tried to reproduce TabDDPM's results, we noticed it performs very well on most datasets but does poorly on the News dataset. This is not due to training issues, since the training curve looks normal. Instead, the generated content converges to repetitive values rather than a diverse distribution. This explains the failure to generate meaningful News content. To illustrate, we show the samples from the News training set and the corresponding TabDDPM syntheses. The training data exhibits variety, while the model's output is stuck in a narrow mode.
>
> Training Set (News dataset)
> |   timedelta | n_tokens_title | n_tokens_content | n_unique_tokens |n_non_stop_words  | ... |
> | --------   | ------- | ------- |  ------- | ------- | ------- |
> |  423.0 | 7.0 | 259.0 |0.6762295 |1.0 |
> |  147.0 | 10.0 | 577.0 | 0.5044092 | 1.0|
> |  665.0 | 8.0 | 252.0 | 0.6468254 | 1.0 |
> |  593.0 | 10.0 | 167.0 | 0.7468354 | 1.0 |
> |  ... | ... | ... | ... | ... |
>
>
> TabDDPM Synthesized Data (News dataset)
> |   timedelta | n_tokens_title | n_tokens_content | n_unique_tokens |n_non_stop_words  | ... |
> | --------   | ------- | ------- |  ------- | ------- | ------- |
> |  731.0 | 23.0 | 0.0 | 701.0 | 1042.0 |
> |  731.0 | 23.0 | 0.0 | 701.0 | 1042.0 |
> |  731.0 | 23.0 | 0.0 | 701.0 | 1042.0 |
> |  731.0 | 23.0 | 0.0 | 701.0 | 1042.0 |
> |  ... | ... | ... | ... | ... |
>
>
> To investigate further, we attempted to remove the two columns of categorical features from News, leaving only numerical features. With this simplified dataset, TabDDPM now synthesizes high-quality diverse examples. It achieved low error rates of 0.81% on Single Column Density Estimation and 1.09% on Column Pair Correlation Estimation. This suggests the categorical columns in News caused issues for TabDDPM in properly learning the true data distribution.
>
> We further investigate the changes in training loss under both scenarios and found that in the absence of categorical columns, the Gaussian noise loss (corresponding to numerical features) can converge to a relatively low value, around 0.25. However, when both categorical columns are present, the Gaussian noise loss can only decrease to around 0.6. This suggests that the multinomial losses from the categorical columns may be hindering the training of the numerical columns. It indicates that the current training method adopted by TabDDPM may have some limitations, although this could be considered a corner case.
>
> > W6: Deploy MLE as a privacy protection metric.
>
> The initial version of the paper used Machine Learning Efficiency (MLE) as a Privacy Protection application because we thought training models on synthetic data instead of original data protected privacy. After listening to this suggestion, we realized that categorizing MLE as a privacy protection task was inappropriate. Therefore, in the updated paper, we have revised this statement.
>
> Additionally, we adopted the suggestion from reviewer dVGm, using the Distance to Closest Records (DCRs) scores to evaluate privacy protection. The corresponding results have been updated in Appendix F.5, Figure 10, and Table 12. The corresponding code updates have also been incorporated into our codebase.

---

> > ### Comment · Reviewer_Fs1y · 2023-11-19
> > **Q2: Sample Size Limitation**
> >
> > Thank you for your diligent response to my inquiries. Upon re-executing the code with a reduced sample size, I experienced no errors, which leaves me puzzled as to the original cause of the issue. However, everything is functioning correctly at present. My score remains unchanged, and I extend my best wishes to the authors. Thank you for your contribution.

---

### Official Review · Reviewer_q7Xg · 2023-11-02

**Soundness:** 3 good
**Presentation:** 3 good
**Contribution:** 3 good
**Rating:** 5
**Confidence:** 3

**Summary:**

This work introduces a latent diffusion model for generating tabular data, and presents a benchmark consisting of six datasets and five quality metrics to evaluate the performance. The comparison in this unified testing environment demonstrates the superiority of the proposed method.

**Strengths:**

•	This paper presents a benchmark that is beneficial to the community.

•	The better performance over previous work across five quality metrics showcases its effectiveness in generating high-quality tabular data.

**Weaknesses:**

1.	The motivation behind using latent diffusion for tabular data generation is not thoroughly discussed in the paper, and the model design does not effectively exploit the characteristics of tabular data.

2.	The VAE decoder design is tailored specifically for either numerical or categorical features, which limits its applicability in a wider range of tabular data scenarios, such as datasets containing a mixture of both numerical and categorical features.

**Questions:**

1.	Are the results shown in Figure 3 derived from the training set or the validation set?

2.	Does replacing the MLP in the diffusion model with a more powerful architecture, such as a Transformer, have any impact on the performance?

---

> ### Author Response · Authors · 2023-11-17
> **Response to Reviewer q7Xg  (1/2)**
>
> We appreciate the valuable feedback and have carefully considered the points raised. Below, we address your concerns, hoping our revisions clarify all the questions and strengthen the quality of our work.
>
> > W1: The motivation behind using latent diffusion for tabular data generation.
>
> We have illustrated the motivation behind using latent diffusion for tabular data in the first paragraph of the Introduction section. To provide a clearer and more concise explanation, we can rephrase it as follows:
>
> One significant distinction between tabular data and image data is the presence of mixed data types in tabular data, including both numerical and categorical features, whereas image data primarily consists of continuous pixel values representing color and intensity.  Standard diffusion processes rely on a continuous input space with Gaussian noise perturbation, making them unsuitable for handling categorical features. Prior work employed different distribution functions for different data types (e.g., Gaussian noise for numerical variables and categorical noise for categorical variables). This approach poses difficulties in the model's ability to effectively capture the co-occurrence patterns among different types of data. To address this limitation and preserve inter-column correlations for tabular data, our approach involves developing a diffusion model in a joint space that accommodates both numerical and categorical features. This choice to use latent diffusion for tabular data generation is driven by the need to bridge this gap.
>
> In the following answer, we illustrate how we achieve this by transforming mixed-type tabular data into a unified space using a well-designed VAE for tabular data.
>
> > W1-2: Model design for tabular data characteristics and handling mixed-type data.
>
> Tabular data have three main characteristics: (1) the mixed type heterogeneous data, including numerical variables and categorical variables, (2) the correlation among columns, and (3) the complex and varied distribution and statistical properties for each column.
>
> - To handle mixed-type data, we have developed specialized feature tokenizers that transform each column into a space with the same *d* dimensions. For numerical columns, we utilize learnable linear transformations, while for categorical columns, we employ learnable embedding lookup tables. Then, these processed features were further fused through a Transformer encoder.
> - To capture correlation among columns, we employ a VAE-based on the Transformer architecture. The self-attention mechanism in Transformer allows each column to dynamically interact and fuse with all others. This is achieved by learning the importance weights of every other column when processing a particular column.
> - To learn the complex tabular data distributions, we combine the VAE and the diffusion process. By training a VAE on tabular data, each column's statistics are compressed into latent variables that capture the essential statistical properties of that column. The VAE is trained using a reconstruction loss function with a KL divergence term. This regularization encourages the latent space to approximate a standard normal distribution, which supports capturing the varied distributions of different columns. After training the VAE, a diffusion process is applied to the latent representations. By needing to denoise the added noises during the diffusion process, the model learns the underlying structure of the data distribution. Together, VAEs and diffusions allow for capturing complex and varied column distributions.

---

> ### Author Response · Authors · 2023-11-17
> **Response to Reviewer q7Xg (2/2)**
>
> > Q1: Are the results shown in Figure 3 derived from the training set or the validation set?
>
> The results shown in Figure 3 are from the validation loss, and we will change the label of the y-axis from "loss" to "validation loss" to make it clearly stated.
>
> > Q2: Impacts of replacing the MLP in the diffusion model with a more powerful architecture.
>
> We experimentally evaluated multiple neural network architectures, including MLPs, MLPs with residual connections, and Transformers for the denoising function (since these structures have been proven effective at modeling tabular data [1]).  Across all architectures tested, we observed negligible differences in denoising performance. Given the faster training and sampling speed of MLPs (detailed in Table 8, Appendix F.1), and for fair comparison to prior work using MLP-based denoising functions, we selected the MLP architecture.
>
>
> The error rates of column-wise density estimation (denoted as Single) and pair-wise column correlation estimation (denoted as Pair) on the Adult dataset for each architecture are presented as follows:
>
> |  Architecture           |  Single            | Pair         |
> | --------                | -------            | -------      |
> | TabSyn (MLP, current design)   |   0.58             |    1.54      |
> | TabSyn (Residual)         |   0.55             |    1.52      |
> | TabSyn (Transformer)    |   0.53             |    1.52      |
>
>
> References:
>
> [1] Gorishniy, Yury, et al. "Revisiting deep learning models for tabular data." Advances in Neural Information Processing Systems 34 (2021): 18932-18943.

---

> ### Author Response · Authors · 2023-11-21
> **A reminder**
>
> Dear reviewer, we have submitted my reply a few days ago. Now the deadline for reviewer-author discussion is approaching, but we observe that you have not replied to my comment. We would be happy if you let me know if you still have some questions or replies. If so, we will reply promptly. Looking forward to your reply.

---

### Meta-Review · Area_Chair_ss2B · 2023-12-10

**Metareview:**

This paper introduces a new method to synthesize tabular datasets with a latent diffusion model.
It natively handles mixed continuous/categorical features by first learning a continuous embedding of all of them to then also exploit interactions between them in a VAE followed by a diffusion model. The method yields very strong results in a comprehensive empirical evaluation. It also makes a contribution to the training algorithm by scheduling the beta of the beta-VAE.
The paper has many strengths, including the inclusion of clean and well-documented source code, a new benchmark with five different evaluation metrics, strong performance in all of these metrics, clarity of writing and clear illustrations, and a single hyperparameter setting used throughout to avoid cherry picking.
Weaknesses identified by the reviewers are that the scientific contribution is somewhat incremental, that it is questionable whether hyperparameter optimization would have helped the baselines more, and the downside of introducing a new benchmark that does not subsume previous benchmarks but might become a competing benchmark (leading to fragmentation and incomparability of results).
Overall, this is a very strong paper. While tabular data has not traditionally been a focus at ICLR (since traditional ML methods have been competitive), I believe it would be good to change this, seeing that there are now many successful applications of deep learning for tabular data. I therefore recommend this paper for an oral.

**Justification For Why Not Higher Score:**

N/A
(One reason not to give an oral could be the relatively incremental methodological contribution.)

**Justification For Why Not Lower Score:**

This paper makes solid contributions, significantly improves the state of the art, and will have substantial input in the community due to its release of a comprehensive benchmark and clean source code for reproducing the paper's results.

---

### Decision · Program_Chairs · 2024-01-16

Accept (oral)